# A self-matched leaky-wave antenna for ultrahigh-field magnetic resonance imaging with low specific absorption rate

G. Solomakha 📧 [1], J. T. Svejda 📧 [2], C. van Leeuwen[3], A. Rennings[2], A. J. Raaijmakers[3,4], S. Glybovski 📧 [1✉] & D. Erni 📧 [2]

The technology of magnetic resonance imaging is developing towards higher magnetic fields to improve resolution and contrast. However, whole-body imaging at 7 T or even higher flux densities remains challenging due to wave interference, tissue inhomogeneities, and high RF power deposition. Nowadays, proper RF excitation of a human body in prostate and cardiac MRI is only possible to achieve by using phased arrays of antennas attached to the body (so-called surface coils). Due to safety concerns, the design of such coils aims at minimization of the local specific absorption rate (SAR), keeping the highest possible RF signal in the region of interest. Most previously demonstrated approaches were based on resonant structures such as e.g. dipoles, capacitively-loaded loops, TEM-line sections. In this study, we show that there is a better compromise between the transmit signal $\mathbf{B}_1^+$ and the local SAR using non-resonant surface coils generating a low electric field in the proximity of their conductors. With this aim, we propose and experimentally demonstrate a leaky-wave antenna implemented as a periodically-slotted microstrip transmission line. Due to its non-resonant radiation, it induces only slightly over half the peak local SAR compared to a state-of-the-art dipole antenna but has the same transmit efficiency in prostate imaging at 7 T. Unlike other antennas for MRI, the leaky-wave antenna does not require to be tuned and matched when placed on a body, which makes it easy-to-use in prostate imaging at 7 T MRI.

[1] Department of Physics and Engineering, ITMO University, St. Petersburg, Russia. [2] General and Theoretical Electrical Engineering (ATE), Faculty of Engineering, University of Duisburg-Essen, and CENIDE – Center for Nanointegration Duisburg-Essen, Duisburg, Germany. [3] Imaging Division, UMC Utrecht, Utrecht, The Netherlands. [4] Department of Biomedical Engineering, Eindhoven University of Technology, Eindhoven, The Netherlands. ✉email: s.glybovski@metalab.ifmo.ru

Currently, there is a clear trend toward higher static magnetic fields $B_0$ in magnetic resonance imaging (MRI) systems. The interest in so-called ultra-high-field scanners, having $B_0$ of 7 T and higher for imaging and spectroscopy of human body tissues, is explained by higher achievable image resolution and contrast with shorter examination times as compared to the widespread 1.5 T and 3 T clinical systems[1]. Thanks to the availability of 7 T whole-body scanners, it is possible to increase our knowledge of biology and medicine by collecting more data, e.g., in the functional investigation of the human brain[2] or of the spinal cord[3]. The 7 T technology even provides insight into mechanisms of neuropsychiatric disorders with much better reliability[4]. However, despite the recent advances in biomedical research using ultra-high-field MRI, this technology is still in the process of its clinical implementation. The main factors preventing clinicians from using ultra-high field MRI for medical diagnostics originate from undesired interference effects of propagating electromagnetic waves in body tissues at relatively high Larmor frequencies. Thus for protons at 7 T, the Larmor frequency increases to 298 MHz so that the average wavelength in body tissues shrinks to 13 cm[5], which is comparable to e.g., the extent of internal organs in the abdominal cavity[6]. Along with the high attenuation of electromagnetic waves due to the relatively high conductivity of the medium, interference effects make the radio frequency (RF) magnetic field intrinsically inhomogeneous[7]. As a result, body images at 7 T inevitably have dark voids[6].

The inhomogeneity issue is typically addressed in research ultra-high field MR systems by the method of parallel transmission (pTx)[8,9]. Unlike birdcage coils used for whole-body RF-excitation at the low frequencies of clinical scanners[10], pTx allows for the manipulation of the transmit field distribution. This is achieved by using multiple surface coils placed directly onto a body and driven with customized phases and amplitudes, which allows steering the signal voids away from the region of interest. However, this approach is still not allowed in clinical MRI as it typically requires careful preliminary determination of individual transmit phases and constancy of coil tuning and matching for each subject. Another limitation is the potentially high peak local specific absorption rate (SAR) created by each antenna element typically due to its proximity to the body surface. At the same time, properly designed transmit surface coils should maximize the signal in the region of interest (ROI) for a given applied power. The signal is proportional to the correlating components of the circularly polarized RF magnetic field $B_1^+$. In the case of abdominal cavity imaging at 7 T (in particular, prostate imaging), the ROI is located one or more wavelengths away from the surface coil and can be considered as an intermediate field region. In this region, the electromagnetic field already resembles a highly attenuated propagating wave rather than a quasi-static field[11]. The best transmit coil should maximize the ratio between $B_1^+$ and the square root of peak local SAR. For body imaging, it means that a coil needs to create as high $B_1^+$ as possible even in the region of interest (ROI) around the centre of the body while minimizing the level of SAR at its hotspot (typically close to the coil). Solving the SAR issue, which is necessary for the potential clinical application of pTx, is strongly related to the design of surface coil elements.

Corresponding transmit surface coils at low frequencies may typically consist of surface loops[9,12] arranged parallel to the body surface. However, since the far-field of a vertical magnetic dipole in its axial direction is zero, surface loops become inefficient at high frequencies. Indeed, at 7 T $B_1^+$-magnetic field produced by a surface loop is strongly inhomogeneous[13,14]. For this reason, alternative types of surface coils have been proposed. Stripline resonators[15–17] allow a more densely packed transmit array without using decoupling circuits. Higher Q-factors of stripline resonators lead to higher currents in the coil and hence, to higher $B_1^+$ for the same transmit power, but only for the quasi-static near field region. In[11] it was shown that deeply located imaging targets in 7 T whole-body imaging are located outside the near field region. To better image such ROIs, coils must be designed to redistribute the magnetic RF energy toward the intermediate- or even far-field region (these are referred to as radiative coils[18]). One of the most efficient types of their types used for transceiver body arrays at 7 T is the dipole[18]. Among several known configurations of dipoles used e. g. for prostate imaging at 7 T, fractionated dipoles[11] have demonstrated the best compromise between the transmit efficiency for deeply located targets and the peak local SAR[19]. A dipole oriented along the $B_0$ field typically creates the maximum RF magnetic field for the given depth of the ROI right under its centre regardless of distance. Although a dipole as a resonator has a much smaller Q-factor compared to a loop, it still radiates due to a standing wave of electric currents in its conductors. In MRI near-half-wavelength dipoles efficiently convert almost all their radiated power into dissipative losses inside the body. Recently, it was demonstrated that two parallel dipoles with in-phase currents combined into the same coil further reduce the peak local SAR as compared to a single dipole due to weaker near fields and lower Q[20]. As alternatives to dipoles and loops, other surface coil types are suitable for operation at 7 T, such as transverse slots[21] and even water-filled slotted waveguide resonators[22].

Noticeably, all previously designed surface coils known from the literature are resonant. The only known non-resonant excitation method, called traveling-wave MRI, uses the cylindrical bore (RF shield of a scanner) as a waveguide in which, at Larmor frequencies larger than 270 MHz, a propagating $TE_{11}$ mode is supported. This mode, typically excited with a patch antenna, delivers the RF signal to a human body or a head[23–25]. Despite providing low SAR, this volume excitation type strongly suffers from low transmit efficiency and weak parallel transmit capabilities. As for efficient surface body coils for pTx, one can find a direct correspondence between the Q-factor of the resonator used in a coil and the peak local SAR induced for the same $B_1^+$ at large depths in a body.

In this study, we propose and experimentally demonstrate a non-resonant surface coil for ultra-high field body imaging with a new RF excitation mechanism and low SAR, in which a wave traveling in slotted microstrip line launches leaky waves into a human body as schematically depicted in Fig. 1. Due to its non-resonant radiation, it induces only slightly over half the peak local SAR compared to a state-of-the-art dipole antenna but has the same transmit efficiency in prostate imaging at 7 T. Unlike other antennas for MRI, the leaky-wave antenna does not require to be tuned and matched when placed on a body in a wide frequency range.

## Results

**Resonant and leaky-wave RF excitation in MRI.** All surface coils, such as, e.g., dipoles, loops, and stripline segments, consist of resonators placed close to a human body. As such, their operation is based on the excitation of standing waves. This approach necessarily results in the excitation of strong reactive electric and magnetic fields in a coil's vicinity. It is well-known in antenna engineering that the higher the Q-factor, the stronger the near reactive fields, i.e., the magnetic fields directly created by currents and electric fields directly created by charges. In ultra-high field body imaging, the magnetic components in the reactive near field are concentrated near the surface coil and, in contrast to radiative fields (produced by displacement currents in the medium), do not contribute to the signal in deeply located ROIs.

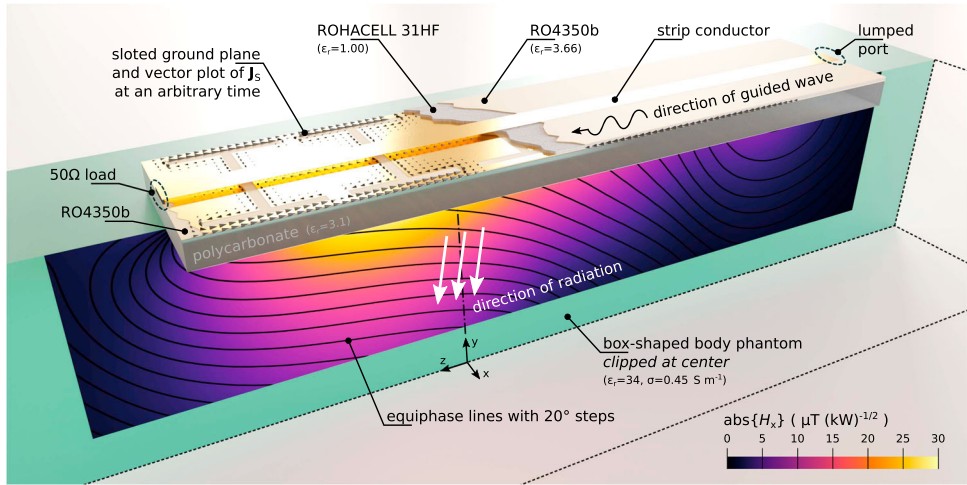

**Fig. 1 Proposed RF excitation mechanism through non-resonant radiation of leaky waves.** The exciting slotted microstrip line radiates into a box-shaped human body phantom for body MRI RF excitation. The color plot shows the normalized magnitude of the simulated $H_x$ distribution, while black contours represent wavefronts in the conductive medium. Calculated surface current density distribution on the slotted ground plane is shown with black arrows.

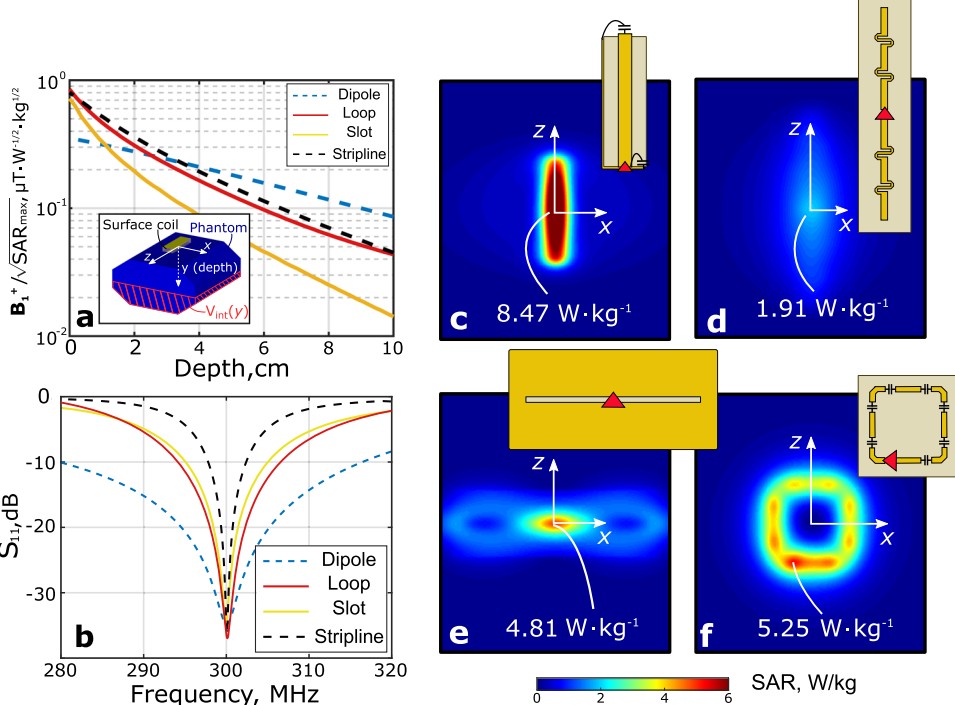

**Fig. 2 Correspondence between the peak local SAR and resonant properties of surface coils for body imaging at 7 T. a** $B_1^+/\sqrt{SAR}$ factor for different depths of ROI; **b** reflection coefficient at the input of a coil; simulated local SAR patterns at 1 W of accepted transmit power at the top surface of the phantom: **c** stripline; **d** fractionated dipole; **e** slot and **f** loop coil.

Therefore, these fields are useless, e.g., in prostate imaging. However, the reactive electric field components play an important role as they may cause local SAR hotspots.

To illustrate the relationship between the peak local SAR and the resonant properties, we have numerically analyzed four of the most popular surface coils used as elements of body transmit arrays at 7 T. In the simulation, the coils were placed on top of a pelvis-shaped body phantom ($\varepsilon_r = 34$ and $\sigma = 0.45$ S/m) with a 1.5-cm-thick spacer. The same properties of a phantom were used throughout the paper. All four coils were tuned at 300 MHz and matched to 50 Ω with appropriate Π-circuits of lumped elements and accept the same power of 1 W from the transmitter. Figure 2 shows the results of the comparison. From field profiles

normalized by the square root of peak SAR (the so-called SAR efficiency) in Fig. 2a and from SAR profiles in Fig. 2c–f, it follows that the compared coil types are very different in terms of RF safety (i.e., they create different SAR levels for the same of accepted power). Figure 2b shows the frequency dependencies of the reflection coefficient $S_{11}$ at the feed port of each coil, which illustrates the bandwidth of the impedance is smaller for larger Q-factor. As shown in Fig. 2a, the stripline coil created the highest SAR for the lowest $B_1^+$ at depths larger than 5 cm, while its bandwidth was the smallest. For the fractionated dipole, the result is the opposite. Comparing the maximal SAR values and bandwidths of the coils, one can clearly check that the broader the bandwidth of the coil, the lower the peak SAR it creates at the

**Table 1 Numerically simulated results of the $B_1^+$ transmit and local SAR efficiencies for different coils placed on a homogeneous human body phantom.**

| Coil | $B_1^+ / \sqrt{P_{acc}}$ $\mu T \cdot W^{-1/2}$ | Simulation $B_1^+ / \sqrt{max(SAR)}$ ($\mu T \cdot W^{-1/2} \cdot kg^{1/2}$) | max(SAR) ($W \cdot kg^{-1}$) | Experiment $B_1^+ / \sqrt{P_{acc}}$ ($\mu T \cdot W^{-1/2}$) | $\Delta T$ (K) |
|---|---|---|---|---|---|
| LWA | 0.26 | 0.22 | **1.42** | 0.27 | 0.16 |
| Dipole | 0.26 | 0.19 | 1.91 | 0.27 | 0.26 |
| Stripline | 0.13 | 0.09 | 8.47 | – | – |
| Loop | 0.07 | 0.05 | 5.25 | – | – |
| Slot | 0.12 | 0.07 | 4.81 | – | – |

The $B_1^+$ field magnitude was taken at a depth of 7 cm.
Best values are highlighted in bold.

phantom surface. The resulting key properties of $B_1^+$ and the maximum of the SAR for these four coils are presented in Table 1.

Concerning the observed correspondence, one can assume that reactive near fields and SAR are related quantities. It is shown in the "Methods" section that the relation is implicitly given by the electric and magnetic fields as they have to fulfill a boundary value problem and hence resulting in corresponding distributions of reactive fields and SAR. However, the Q-factor is an integral parameter and dependent on several factors. Additionally, the Q-factor cannot be derived by the bandwidth of the resonance in the case of non-resonant antennas. Regarding these facts, a second measure of reactive behavior of a coil, namely the difference of electric and magnetic energy densities $\Delta w$, is utilized to provide more insight into the reactive near fields and their extent in the region of the phantom.

The difference of energy densities $\Delta w$ is a local and directly related measure for reactive fields as it vanishes in lossless regions permeated by propagating wave fields, whereas a significant difference occurs in regions of dominant reactive near fields, typically around resonating structures, due to the imposed spatial separation of electric and magnetic energy densities. In lossy media, represented here by the permittivity and the conductivity of the human body tissue, the difference $\Delta w$ remains nonzero even for the latter case of a propagating wave due to induced (eddy) currents. This attenuated wave addresses, therefore, the least resonant state in the lossy medium yielding the smallest possible difference $\Delta w$, which is thus associated with a minimal reactive field.

In order to visualize the presence of reactive fields created by the different coil types inside the phantom, the following comprehensive measure $\xi$ is now introduced. Given the definition

$$\xi(y) = \int_{V_{int}(y)} 2\omega |\Delta w| dV \quad (1)$$

where the absolute value of $\Delta w$ is integrated over the $y$-dependent volume $V_{int}$, which covers the whole phantom from the outermost boundary at $y = d_y$ up to a given horizontal cross-section at position $y$. The actual volume of integration is indicated at the inset of Fig. 2a.

Therefore, $\xi(y)$ is perfectly apt to visualize the spatial behavior of the (horizontally averaged) reactive near-fields around the coil in relation to the best case of an attenuated propagating wave field as a function of the penetration depth $y$ into the phantom, which is shown in Fig. 3 for all four coil types.

As can be seen, in terms of the evaluated spatial separation of electric and magnetic fields, the loop coil having the highest value for $\xi$ is the most reactive among four above compared coils, while the dipole is the most radiative having the lowest $\xi$. Below it is

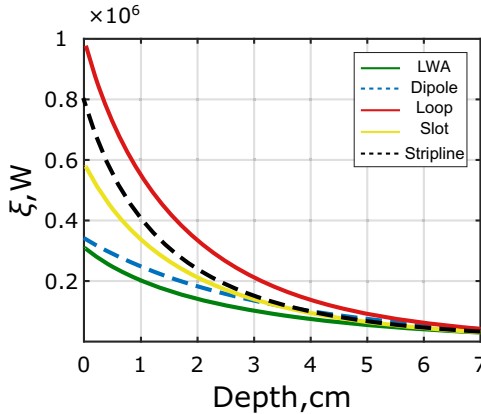

**Fig. 3 Comparison of different MRI coil types according to their reactive near fields.** Figure of merit based on the difference between electric and magnetic energy density as defined in equation (1) for different coil types equally fed with 1 W of accepted power as a function of depth in the phantom.

demonstrated that a completely non-resonant coil has even lower $\xi$ and lower SAR.

To design a surface coil with an even lower reactive field and SAR, we proposed to use a non-resonant structure operating as a leaky-wave artificial transmission line. The idea is illustrated in Fig. 1, while the operating principle is similar to the ones of leaky-wave antennas (LWAs), typically used in the microwave range and above. In such antennas, a structure guiding a propagating wave continuously radiates into free space. According to ref. [26], one of the main types of LWAs is called the uniform LWA, in which the guiding transmission line is uniform along its length and supports a wave that propagates faster than the speed of light in free space. In other words, the phase constant $\beta$ in the complex propagation constant $k_z = \beta - j\alpha$ stays in the range $0 < \beta < k_0$, where $k_0$ is the free-space wavenumber. Here we propose to use a periodic structure of slots with a sub-wavelength periodicity as an artificial transmission line with the possibility to adjust the wave dispersion[27], so it can be classified as a quasi-uniform LWA.

In LWAs the phase constant $\beta$ is adjusted to define the radiation angle $\theta_{rad}$ through the approximate relation $\sin \theta_{rad} = \beta/k_0$ at given frequencies within the whole hemisphere[28]. At the same time, the attenuation factor (leakage constant) $\alpha$ of the guided wave is chosen to maximize the gain by radiating as much of the applied power as possible along the propagation length $L$[29]. As long as the propagating wave loses most of its power in the form of radiation before reaching the far end of the transmission line, the antenna stays non-resonant, and its input impedance

remains relatively constant with frequency. To avoid reflections from the end, matched loads or power recycling networks[30] are connected to the end of the transmission line. Usually, LWAs are designed to radiate more than 80 % of its guided input power. In this case, the terminal load has a negligible influence on the LWA performance. However, to avoid standing waves, the end load should be matched to the line.

To radiate to the free space, LWAs should support the propagation of a fast wave[26]. In contrast, surface coils for MRI should operate when positioned close to a human body, which represents a medium with high conductivity and high permittivity. In this case, leaky-wave radiation is allowed even if $\beta$ is comparable to $k_0$. Indeed, as it has been previously demonstrated, even a simple slot line, which cannot radiate when situated in free space, becomes an efficient leaky-wave radiator when place at a boundary of a half-space with high permittivity[31]. Leaky-wave transmission lines have been used to radiate into high-permittivity media microwave lens antennas[32], in ground penetrating radars[33], and as applicators for microwave medical imaging[34,35]. When speaking of radiation beneath the boundaries of conductive media, leaky-wave applicators have been demonstrated as appropriate sources of inhomogeneous (exponentially decaying) waves, which can cause the effect of deep electromagnetic penetration into a conductive medium[36,37]. Along with the non-resonant properties of leaky-wave radiation with potentially lower near fields, the possibility to exploit the deep penetration effect to reach high transmit efficiency in deeply located ROIs was our motivation to propose a leaky-wave antenna for MRI.

To achieve the leaky-wave excitation at the Larmor frequency of protons at 7 T (around 300 MHz), we designed our antenna as a microstrip-line section with a periodically slotted ground plane. The guided wave in this microstrip line is designed to be faster than the plane wave in the medium of the phantom, leading to leaky-wave radiation into the lossy phantom and consequent radiated power absorption. The transmission line with a strip

width of 15 mm and height of 2 mm is matched to the transmitter at its input port. The ground plane of the microstrip is separated from the strip by a 2-mm-thick foam layer and has six identical I-shaped slots that are periodically arranged in the $z$-direction (along the static field $\mathbf{B_0}$ in MRI) with the period of 60 mm. The slots in this microstrip line are required to increase the radiated power per unit length of the line. The slot length $L_s$ affects both the phase constant $\beta$ and the leakage constant $\alpha$. As with any LWA, the proposed LWA structure delivers some residual power to the far end of the transmission line, that is supposed to be fully absorbed by the matched load to avoid standing waves. To isolate the slotted ground plane of the antenna from the phantom, a 1.5-cm-thick polycarbonate spacer was used.

**Design and optimization of the LWA.** We designed the leaky-wave antenna for prostate imaging at 7 T, as an element of a pTx array. Therefore, we limited its length and width to 40 cm and 8.5 cm, respectively, in order to be able to accommodate eight such antenna elements around a human pelvis. In fact, we used the maximum available length to maximize the propagation length of the exponentially decaying wave along the microstrip line in order to maximize power radiation before terminal load. The same reasoning applies to the design of the slot geometry.

To obtain a sufficient leakage factor $\alpha$, the I-shape slots in the ground plane have been arranged with the period of $p = 6.4$ cm along the microstrip line. The length and number of the slots in a single antenna element were optimized to maximize the $\mathbf{B_1^+}$ level at a depth of the prostate for a given transmit power, as described in the "Methods" section. With this aim, the frequency dispersion of both $\beta$ and $\alpha$ in the proposed leaky-wave transmission line were parametrically studied by numerical simulation of the single unit cell depicted in Fig. 4a. The results depending on the slot length $L_s$ are shown in Fig. 4. From the dispersion curves given in Fig. 4c in the presence of the phantom and in Fig. 4d without the phantom, it is seen that increasing $L_s$

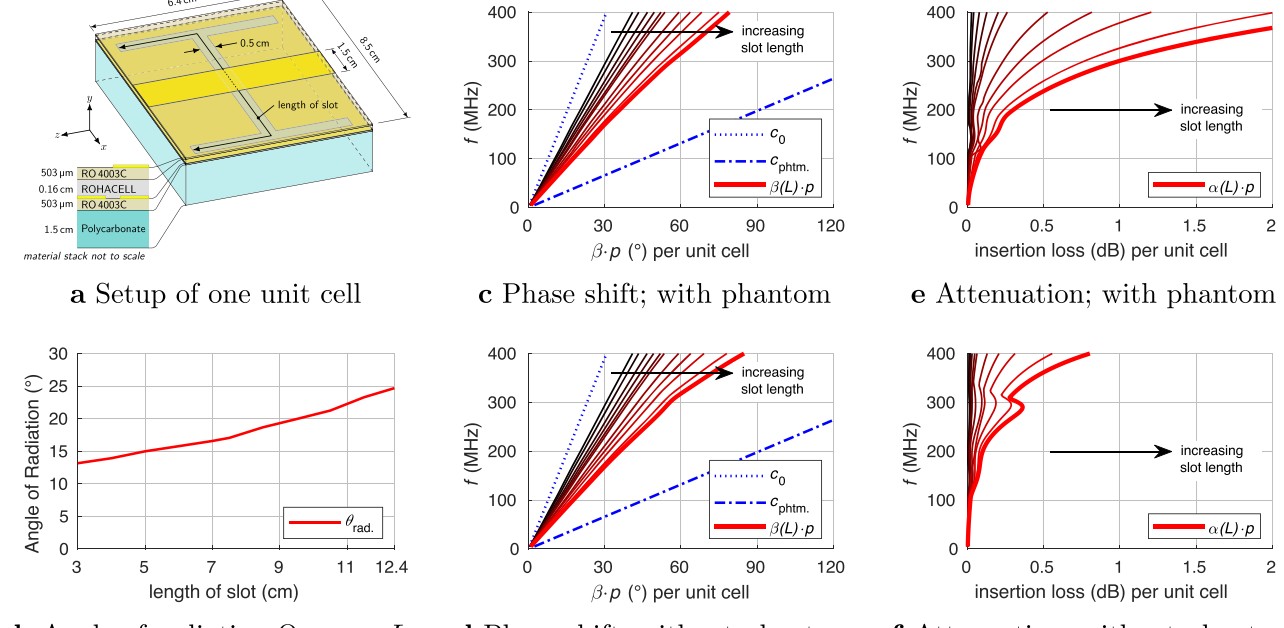

**a** Setup of one unit cell

**b** Angle of radiation $\Theta_{\mathrm{rad}}$ vs. $L_s$

**c** Phase shift; with phantom

**d** Phase shift; without phantom

**e** Attenuation; with phantom

**f** Attenuation; without phantom

**Fig. 4 A parametric numerical investigation of dispersion characteristics based on single unit cell.** Geometry of the unit cell (**a**), angle $\theta_{\mathrm{rad}}$ of the leaky-wave radiation for the loaded transmission line (**b**), phase shift ($\beta \cdot p$) and insertion loss (dB) ($\alpha \cdot p \cdot 20 \log{(10)}^{-1}$) in the microstrip leaky-wave transmission line per unit cell (**c**, **d–f**). The length of the slot is varied from 3 cm to 12.4 cm. In cases **c**, **e**, the transmission line is placed on a phantom while cases **d**, **f** represent the free space environment of the transmission line. The bold curves are for the LWA with an optimal length of 12.4 cm.

slows down the propagating wave. The dispersion curve, in this case, remains below the light line for free space (with the phase velocity of $c_0$), so that leaky-wave radiation into free space is inhibited. With respect to the propagation conditions in the phantom medium, the same waves act as fast waves where all associated dispersion curves remain above the corresponding light line (with the phase velocity of $c_{phtm}$) yielding leaky-wave radiation into the phantom.

Note that the presence of the phantom has almost no effect on the phase constant $\beta$ (compare Fig. 4c and d). Conversely, as follows from the comparison of Fig. 4e and f, the leakage factor $\alpha$ is strongly increased in the presence of the phantom due to the leaky-wave radiation. It can be seen that $\alpha$ can be continuously adjusted by a corresponding choice of $L_s$. Having $\alpha$ below a particular limit results in lower power efficiency due to dissipation losses in the end load, which decreases the signal level in the ROI. A value of $\alpha$ above another limit leads to a very inhomogeneous field pattern in the lateral direction in the phantom and again a decrease in the ROI. The slot length $L_s = 12.4$ cm was chosen to obtain an attenuation per unit cell of $-1$ dB. In this case, the antenna configuration with $N = 6$ slots yields a residual power at the far end of the line of around -6 dB relative to the input power. The corresponding phase constant of the optimized antenna $\beta \approx 15.21$ m$^{-1}$ leads to a theoretical radiation angle of $\theta_{rad} \approx 22.76°$ (cf. equation (2) in the "Methods" section).

For the given length of the antenna, the chosen thickness results from the optimal trade-off between $\mathbf{B}_1^+$ in the ROI and the peak local SAR. In fact, for thicknesses less than 1.5 cm the level of SAR of the leaky-wave antenna is worse than compared to a dipole antenna with the spacer thickness of 1.5 cm with equal levels of $\mathbf{B}_1^+$ in the ROI.

This optimized configuration makes the LWA best suitable for prostate imaging in terms of power efficiency. In the next subsection, the optimized LWA is numerically and experimentally compared to a state-of-the-art antennas for prostate imaging at 7 T.

**Leaky-wave antennal vs. resonant dipole in prostate imaging.** Prostate imaging is one of the most challenging tasks in whole-body MRI at 7 T. The main challenge in designing surface coils for this application is to maximize the magnitude of $\mathbf{B}_1^+$ in the prostate region, which is at a depth of approximately 8 cm (comparable to a wavelength), while keeping the peak local SAR close to the body surface as low as possible. For comparison, we took a fractionated dipole of length 30 cm proposed in[19] (see inset in Fig. 2d), which is the state-of-the-art radiative and resonant array element for the application.

In the simulations of the LWA placed over the homogeneous octagonal pelvis-shaped phantom[19], both terminals of the micro-strip line, namely at the feed and at the far end are matched to 50 $\Omega$. The computed reflection coefficient $S_{11}$ at a feed port is shown in Fig. 5a for both antennas. For the LWA, it is below $-12$ dB in the whole frequency range of 200–400 MHz, whereas for the dipole it remains below $-12$ dB within a 30 MHz band around the central frequency of 298 MHz. The approximate relative amount of radiated power $\nu = 1 - |S_{11}|^2 - |S_{12}|^2$ absorbed by the phantom is shown in Fig. 5b demonstrating the expected non-resonant behavior of the LWA. The value $\nu$ includes the absorbed power in the phantom together with the losses due to radiation into free space and the losses dissipated inside the antenna. However, as follows from numerical simulations, the latter two contributions remain lower than 7%. Therefore, the value of $\nu$ approximately equals the power absorbed in the phantom, which is around 60% and matches up well with the numerically predicted value from the design stage based on the selected $\alpha$.

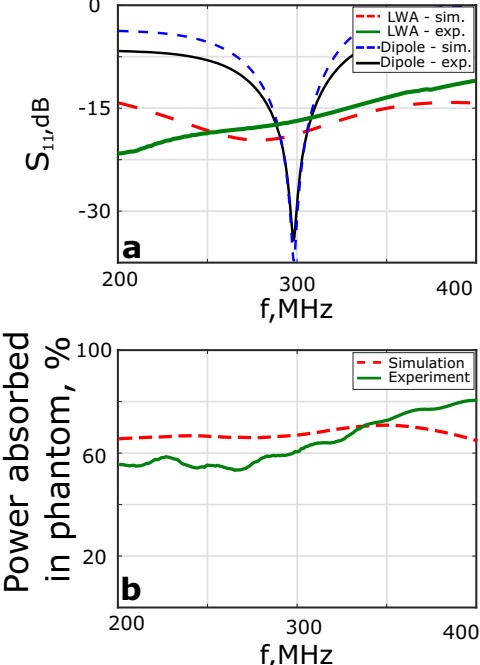

**Fig. 5 Simulated and measured properties of the optimized LWA and the reference dipole antenna. a** $|S_{11}|$ at the feed port and **b** relative power $\nu$ radiated into and absorbed by the phantom.

The simulated $\mathbf{B}_1^+$ distributions in the $YZ$-plane of the phantom (see inset in Fig. 6) for the leaky-wave antennas and the dipole antennas are presented in Fig. 6a and b, respectively. Both antennas were simulated with an accepted input power of 1 W and provide almost the same $\mathbf{B}_1^+$ signal level of 0.27 µT in the ROI at the depth corresponding to the prostate location in a human body (indicated by the ellipses in Fig. 6). It can be verified that the two antennas yield different longitudinal positions for the maximum $\mathbf{B}_1^+$ with respect to the corresponding antennas centers. While the pattern formed by the dipole is symmetric, the LWA forms field pattern with a narrower maximum shifted toward its far end.

The calculated $\mathbf{B}_1^+$ vs. depth profiles taken along dashed lines and going through the centers of the ROIs as indicated in Fig. 6a and b are compared in Fig. 7a. In the vicinity of the surface, the flux densities $\mathbf{B}_1^+$ of the dipole antennas are larger, but starting from a depth of 5 cm (including ROI), both antennas create the same field level. Based on the numerical simulations, we computed the figure of merit, namely the distribution of $\xi$, where its non-vanishing value indicates the spatial extent of a resonant near field as depicted in Fig. 3. Compared to the dipole antenna and the other reference coils, the LWA creates the lowest reactive power contribution at every depth.

The leaky-wave radiation mechanism is clearly illustrated by the simulated phase patterns of the transmit magnetic field created by the LWA in the $YZ$-plane of the phantom. Unlike the dipole antenna, which creates horizontal phase fronts (see Fig. 7c), the phase fronts of the LWA produce a frequency-dependent angle with respect to the $y$-axis. This effect is similar to frequency beam steering by LWAs in free space. The behavior of phase fronts at 250 MHz, 298 MHz, and 350 MHz is shown in Fig. 7d–f.

The simulated SAR at an input power of 1 W, distributed in the top surface of the phantom where the coil is placed ($XZ$-plane according to Fig. 2a), is shown in Fig. 6. The simulated $\mathbf{B}_1^+$ distribution in the ROI and the maximum of local SAR for the LWA, the dipole antenna, and the other reference coils are

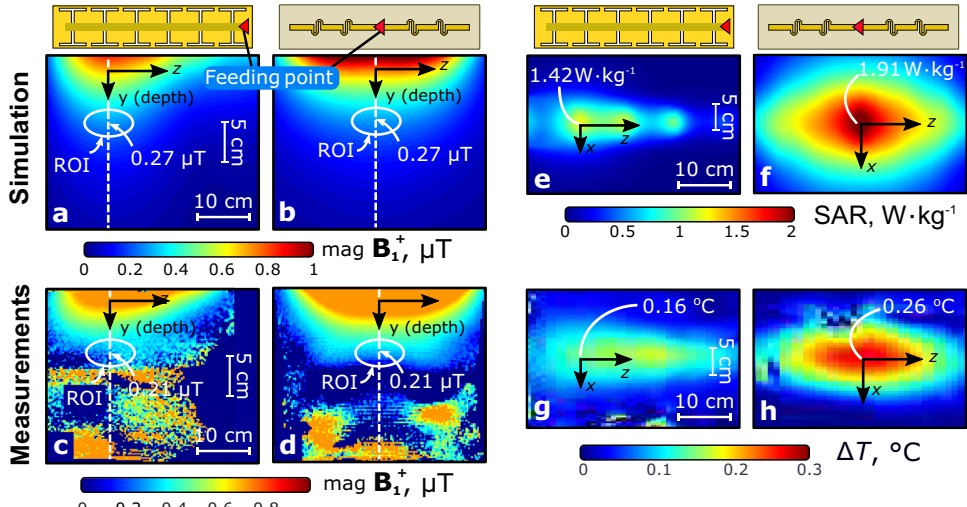

**Fig. 6 Simulated and measured field distributions for 1 W of accepted input transmit power in the octagonal phantom.** $\mathbf{B}_1^+$ in the YZ-plane of the phantom for the LWA (**a**, **c**) and fractionated dipole antenna (**b**, **d**); local SAR and $\Delta T$ in the top plane (XZ) for LWA (**e**, **g**) and dipole antenna (**f**, **h**). The feeding point of the antennas is indicated with a red triangle.

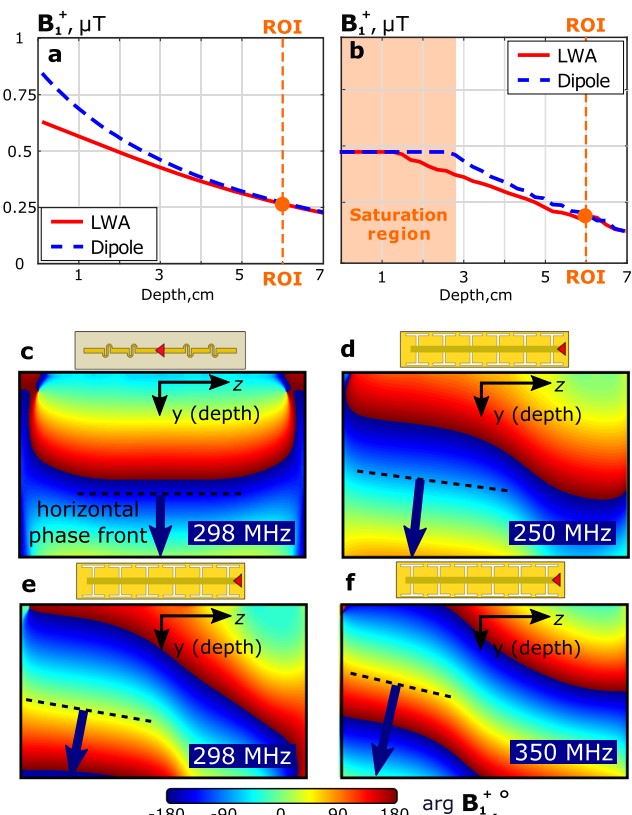

**Fig. 7 Numerical and experimental $\mathbf{B}_1^+$ magnitude and phase profiles.** Simulated (**a**) and measured (**b**) $\mathbf{B}_1^+$ vs. depth profiles for the LWA and fractionated dipole antenna element in the phantom (profiles were plotted along the corresponding dashed lines shown in Fig. 6a–d); simulated phase patterns for $\mathbf{B}_1^+$ in the phantom for the dipole antenna at 298 MHz (**c**), and for the LWA at 250 MHz (**d**), 298 MHz (**e**) and 350 MHz (**f**). The tangent to the phase front for each phase pattern is indicated with a black dashed line.

depicted in Fig. 2. From this comparison, one can see that indeed, the fractionated dipole creates lower maximum local SAR than conventional reference coils, while the LWA has even 27% lower peak SAR value compared to the dipole antenna. Note that

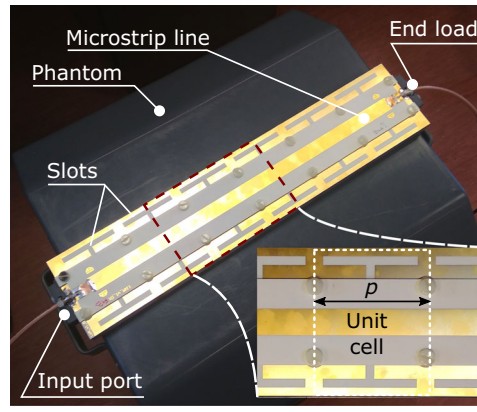

**Fig. 8 Constructed protype of leaky-wave antenna.** Photograps of the fabricated prototype of the optimized LWA placed over a pelvis-shaped homogeneous phantom for MRI characterization.

LWA provides the same $\mathbf{B}_1^+$ in the ROI as the dipole antenna for the same input power.

For the experimental comparison, both the proposed and the reference dipole antennas were manufactured. Four antennas of each type were built to set up a four-element transmit array configuration for in-vivo imaging. A manufactured LWA placed on the phantom is displayed in Fig. 8. The frequency responses of $S_{11}$ and the relative power $v$ radiated into a subject for the proposed LWA and dipole antenna are compared with corresponding simulations and displayed in Fig. 5a and b. In the simulation the antenna was placed on the phantom, while in the experiment it was placed on the body of a volunteer.

The measured $\mathbf{B}_1^+$ field distribution are deduced from flip angle measurements on the Philips Achieva 7 T MRI scanner and displayed in Fig. 6c and d in the same plane as for the previously given numerical results. The measured $\mathbf{B}_1^+$ vs. depth profiles are compared for both antennas in Fig. 7b. In this figure, the orange area (depths smaller than 5 cm) corresponds to a saturation of the measured $\mathbf{B}_1^+$ level where the field levels cannot be compared. The field level changes dramatically from the surface to the center of the phantom due to attenuation. Therefore to carefully compare the $\mathbf{B}_1^+$ levels close to the ROI within the dynamic range of the $\mathbf{B}_1^+$

mapping method, we had to over-saturate the field level measured at depths smaller than 3 cm. However, for depths larger than 5 cm, the measured profiles are very similar to the simulated ones and confirm that both antennas provide the same penetration of the transmit field. The artifacts visible in the measured figures below the ROI, are due to a hole through the phantom and insufficient signal-to-noise ratio for $B_1^+$ mapping.

In support of a validation of the simulated SAR distributions, temperature measurements in the corresponding plane of the phantom (top coronal plane) were measured by MR thermometry. Resulting distributions of temperature increase $\Delta T$, normalized to the accepted power, are presented in Fig. 6g and h. For the LWA, the temperature increase for 1 W of accepted power reached 0.16 °C at its maximum, while for the dipole the maximum was 0.26 °C for the same accepted power, i.e., our LWA yields a 38 % lower temperature increase for the same input power.

For the in-vivo evaluation of a four-element array, local SAR and $B_1^+$ distributions in the detailed high-resolution anatomical models Duke from the virtual population (ITIS Foundation, Zurich, Switzerland) and corresponding Gustav from CST voxel family (CST, Darmstadt, Germany) were simulated for an input power of 1 W. A transverse slice of the Duke voxel model through the prostate is shown in Fig. 9 for the LWA array (a) and dipole array (b), respectively, whereas the associated SAR distributions are depicted in Fig. 9c and d.

For all considered coil types, four-channel array configurations have been simulated on the Duke anatomical model. The resulting key properties of $B_1^+$ averaged over the ROI, and the maximum of the SAR averaged over a volume of 10 g equivalent mass are given in Table 2. An additional comparison was made between the LWA array and the dipole array on the Gustav anatomical model.

It is seen that the LWA array creates a 41 % lower peak local SAR compared to the dipole array for the same input power, and 7% higher $B_1^+$-field averaged over the ROI for the same amount of stimulated power for the Duke anatomical model. Within the Gustav anatomical model, the LWA array creates a 22 % lower peak local SAR compared to the dipole array and almost the same $B_1^+$-field averaged over the ROI.

To demonstrate the performance of the proposed LWA in the application, an array of four identical elements used in transceive mode for prostate imaging of a healthy volunteer. The leaky-wave antennas were tightly placed around the body of the volunteer at two locations on the back and two on the stomach. Field distributions of $B_1^+$ for the LWA array and the dipole array are shown in Fig. 9e and f. From the results one can observe that LWA array creates 14% lower $B_1^+$-field averaged over the ROI. The obtained $T_1$-weighted MR image applying the proposed LWA array and the dedicated reference dipole array in the transverse plane going through the center of the prostate is shown in Fig. 9g and h.

## Discussion

In this study, we numerically and experimentally investigated the direct correspondence between the resonant properties of surface body coils for MRI with the peak local SAR that they create close

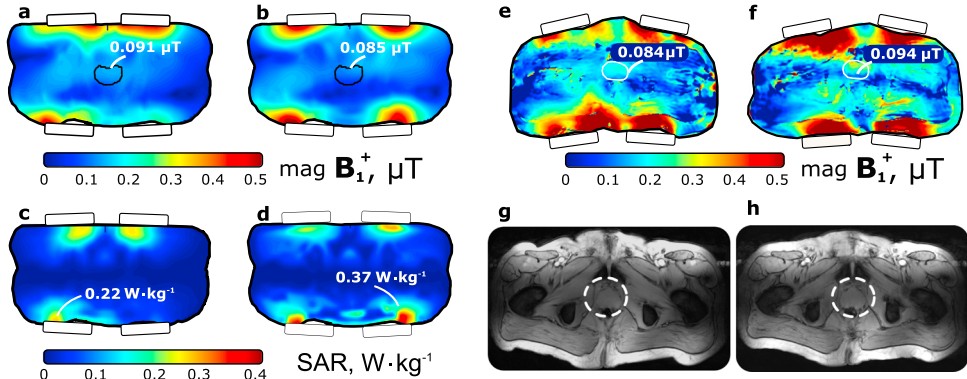

**Fig. 9 In-vivo and safety characterization of four-element LWA array.** Simulated $B_1^+$ for the four-element LWA array configuration (**a**) and a fractionated dipole array (**b**) in the transverse slice through the prostate of the human body Duke anatomical model for 1 W of accepted power. The corresponding SAR for the LWA array (**c**) and fractionated dipole array (**d**) in a transverse slice through the maximum of local SAR. In-vivo $B_1^+$ map in healthy volunteer for LWA array (**e**) and the dipole array (**f**). **g** In-vivo $T_1$-weighted MR image (transverse slice through the prostate) of a healthy volunteer obtained using a four-element array of LWAs and **h** of dipoles.

**Table 2 Numerically simulated results of $B_1^+$ transmit and averaged 10 g-averaged SAR efficiencies for different coils in four-channel configurations placed around two different human body voxel models.**

| Coil array | Body model | $\langle B_1^+ \rangle_{ROI}/\sqrt{P_{acc}}$ ($\mu T \cdot W^{-1/2}$) | $\langle B_1^+ \rangle_{ROI}/\sqrt{\max(SAR_{10g})}$ ($\mu T \cdot W^{-1/2} \cdot kg^{1/2}$) | $\max(SAR_{10g})$ (W/kg) |
|---|---|---|---|---|
| LWA | Duke | 0.091 | **0.193** | **0.22** |
| Dipole | Duke | 0.085 | **0.142** | **0.37** |
| Stripline | Duke | 0.068 | 0.075 | 0.81 |
| Loop | Duke | 0.081 | 0.092 | 0.77 |
| Slot | Duke | 0.054 | 0.049 | 1.1 |
| LWA | Gustav | 0.105 | **0.214** | **0.24** |
| Dipole | Gustav | 0.102 | **0.185** | **0.31** |

$B_1^+$ is averaged over the ROI, whereas the 10 g-averaged SAR is given in the maximum hotspot.
Values to be compared are highlighted in bold.

to the surface of an investigated subject. Firstly this was shown by analyzing the RF-fields created by several coil types, which are the most popular for body imaging at 7 T, i.e., a stripline resonator, a fractionated dipole antenna, a slot antenna, and a loop coil. The comparison shown in Fig. 2 clearly demonstrate that the broader the impedance matching bandwidth (i.e., the lower the Q-factor), the lower the peak local SAR close the surface of a homogeneous body phantom for the same level of $\mathbf{B}_1^+$ at a given depth. Among the compared coils, the fractionated dipole was the best one in terms of the SAR-efficiency, the ratio of $\mathbf{B}_1^+$ at the depth of the prostate and the square root of the peak local SAR (at the hot-spot). The same trend is supported by the simulation of a human body model, as presented in Table 2. So we used the dipole as a reference antenna in the experimental comparison, including the in-vivo MRI scans.

In case of ultra-high field MRI of the prostate, the ROI is located at a depth comparable to the wavelength in the body tissues which corresponds to the intermediate-to-farfield region. At the same time, the hotspot of the local SAR is typically located near the surface of the body, which is inside the near-field region. Human body tissues represent a highly-conductive medium, which yields both, a complex propagation constant and complex characteristic impedance. As a result, RF-fields cannot be easily decomposed into quasi-static and radiation components as in free space. However, as one can see from Fig. 2, the weak electric field in the near field region of the coil with the largest bandwidth, namely the dipole, causes the lowest SAR and the highest radiative magnetic field at large depths among the tested coils. Moreover, from Fig. 3, it is evident that the dipole has the lowest value of $\xi$ at any depth, which indicates low levels of resonant near fields. This leads to the conclusion that a reduction of the peak local SAR, which is the main limiting factor for any transmit coil for ultra-high field body imaging, requires the reactive electric field to be minimized and the bandwidth to be increased. We acknowledge this trend by introducing and demonstrating the first non-resonant surface coil for MRI, the leaky-wave antenna.

Our LWA is based on the continuous energy leakage mechanism and employs no standing waves. By propagating from one end of a microstrip to the other, most of the power leaks into the subject and is absorbed within. The discrete slots of the ground plane being excited by the propagating wave radiate power to the subject and behave as a slot array with almost linear phase spatial variation. As a result, a beam of the field at large depths is produced at an angle that changes with frequency, as seen from the simulations in Fig. 7d–f. The existence of this non-resonant radiation mechanism was simulated and experimentally validated having a good agreement between simulated and measured field patterns and S-parameters of the LWA element.

The study of the transmit signal field $\mathbf{B}_1^+$ in a homogeneous phantom shows that the proposed LWA element creates almost the same level of the magnetic field at a depth of prostate as a state-of-the-art fractionated dipole antennna. This result obtained from simulations on a phantom was precisely confirmed by measurements. However, as shown by temperature increment measurements in the same phantom, the LWA causes just above half of the heating in the near field region than the dipole antenna, which is in good agreement with local SAR simulations. This means that the LWA is more efficient as a transmit antenna than the dipole due to lower reactive near fields, and even more efficient than the other considered coils. In fact, in the vicinity of the surface of the phantom, both the electric and magnetic fields are reduced for the proposed LWA, as was additionally supported by the results of Fig. 3. However, for a depth larger than 3 cm (like the prostate location in the human body), the $\mathbf{B}_1^+$ levels of both antennas at a given transmit power are the same. The advantage in SAR performance has been verified by running

simulations of a four-element array of LWA placed on Duke and Gustav anatomical models. While for both voxel models the level of $\mathbf{B}_1^+$ was close to the ones of the dipole elements with a relative deviation 10%, the difference in local maximum SAR, created by LWA elements, were lower from 41% (for Duke anatomical model) to 30% (for Gustav anatomical model).

Apart from reduced local reactive near fields and smaller peak SAR, the proposed radiation mechanism has the advantage of providing an inherent broadband impedance matching capability. The arrangement of non-resonant slots couples the wave propagating in the microstrip line to the wave propagating in the medium of body tissues almost without any reflection from the far end of the LWA. As a result, the coils provided a return loss of less than −12 dB over the entire frequency band of 200 –400 MHz without the need of a matching circuit.

Furthermore, the input impedance of the LWA in the experiment turned out to be insensitive with respect to any variation of the subject's properties. There was no significant change in the input impedance value when the phantom was replaced by the body of a volunteer, as shown in Fig. 5. In-vivo measurements with four-channel array configurations have shown, however, that the efficiency of the LWA array is 14% less in comparison to the dipole array, although the MR images obtained are very similar for both antenna types.

From an engineering point of view, another advantage of the leaky-wave antenna is that it operates using a radiative microstrip transmission line with an inherently a symmetric topology. Hence no balun network is required when fed by a coaxial line.

Despite the advantages of being self-matched and the lower SAR levels, the LWA also shows some disadvantages. First, the current LWA configuration is relatively long. In order to reduce the length, one could use lumped elements or high-permittivity ceramic material as a spacer material to reduce the slot size and thus the size of the unit cell in future designs. Second, the loading and hence the ratio of $\mathbf{B}_1^+$ to the accepted power of the LWA reaches the one of the dipole only if the LWA is tightly attached to a surface of human body, whereas the dipole is relatively insensitive to an air gap between itself and the surface of the human body. One possibility to address this problem is the method of power-recycling, which is well-known for microwave LWAs[30]. For LWA using this method, one could reroute the residual power back to the feed to increase efficiency in the case of insufficient loading by the subject and where basically the not radiated power is rerouted back to the feeding port increasing the radiation efficiency.

In summary, we proposed and demonstrated a non-resonant surface coil based on the leaky-wave radiation mechanism. It was clearly shown within simulations and experiments that due to lower near electric fields, it reduces peak SAR to just over a half for nearly the same transmit field in the ROI.

Moreover, the LWA does not require either a matching network nor a balun. This allows us to conclude that non-resonant surface coils for ultra-high field MRI, and particularly based on leaky-wave antennas, are very attractive for future research applications of ultra-high field MRI.

## Methods

**Optimization and simulation**. For the numerical analysis and optimization of unit cells (i.e., the slots in the microstrip line), the FDTD-based solver EMPIRE-XPU-2018 (IMST, Kamp-Lintfort, Germany) was utilized. The propagation characteristics were calculated by extracted ABCD-parameters of one unit cell (see Fig. 4) embedded in a fitting microstrip line, i. e. a continuous microstrip line of the same cross-section, but without slots, where the port reference planes are positioned at the terminals of the unit cell[38,39].

The optimal unit cell balances the residual power loss at the end of the line and the radiated field pattern. Short slots lead to weak radiation and high losses in the far end's terminal load and hence to small $\alpha$. The large length $L_s$ of the slots makes them close to their self-resonance, which increases the leakage factor $\alpha$, but it also

changes the angle of radiation (see Fig. 4) by changing the phase constant $\beta$. Large angles $\theta_{rad}$ with respect to the y-axis (broadside direction) lead to distortions of the magnetic field pattern and lower $\mathbf{B}_1^+$ for the same input power. As a result, based on the goal to maximize $\mathbf{B}_1^+$ in the ROI, an appropriate value of $L_s$ and $\alpha$ are first selected from the calculated set of dispersion curves (see Fig. 4e). It is worth noting that the I-shaped slots were chosen to fit the maximum possible width of the antenna (8.5 cm), which allows the accommodation of 8 such antennas around a human body to be able to use an eight-element pTx array configuration in the future.

Next, the corresponding $\beta$ of the dispersive transmission line can be found from the set of curves for $\beta$ in Fig. 4c to estimate the radiation angle (the radiation direction refers here to the normal vectors of planes with constant phase values). The radiation angle $\theta_{rad}$ measured against the broad side direction (y-axis) in the dissipative medium of the phantom is estimated using the following relation based on[36,38]:

$$\theta_{rad} = \arcsin\left(\frac{\beta}{\beta_{phantom}}\right), \tag{2}$$

where $\beta_{phantom}$ equals the imaginary part of the complex wavenumber constant within the phantom.

However, the angle is approximate, as the structure of the LWA is not only finite in length but also comparably short by utilizing only six unit cells. The effect of this shortness on the angle of radiation $\theta_{rad}$ can easily be observed by the phase distribution of the $\mathbf{B}_1^+$-field in Fig. 7, where a radiation angle is hardly deducible.

Simulations of the proposed LWA and the reference dipole antenna in the presence of a phantom and a human body anatomical model were performed in CST Microwave Studio 2019 (CST, Darmstadt, Germany). For simulations with a phantom, adaptive tetrahedral meshing was performed at the Larmor frequency of 298 MHz using the finite-element method (Frequency Domain Solver). The number of meshcells was ~700,000 for all simulated models.

For the safety assessment, four-element arrays of different antenna types were simulated with the same software on the human body Duke anatomical model with the finite integration method. To compare SAR and efficiency of different antenna types, arrays of four identical loops, striplines, and slots with Duke anatomical model were simulated. To prove the stability of SAR reduction the dipole and leaky-wave arrays were also simulated on Gustav anatomical model with different BMI (body mass index). The number of meshcells was ~30 × 10⁶ for all simulated models.

Phase-only static RF shimming, based on $\mathbf{B}_1^+$ phases induced by each of four array elements[40], was applied to maximize $\mathbf{B}_1^+$ in the prostate region. First, the $\mathbf{B}_1^+$ in the ROI was exported to Matlab 2016 (Natick, Massachusetts, USA) for each channel, whereafter, the individual phases of excitation were determined and applied with the same magnitude.

**Antenna prototyping and on-bench measurements**. Each prototype of the LWA was composed of two Rogers RO4003 ($\varepsilon_r = 3.38$, $\tan \delta = 0.003$) 0.203-mm-thick single-layer PCBs, separated by a 2-mm-thick foam layer of ROHACELL 31HF. The top PCB was used to form the stripline, while the bottom one formed the ground plane with 6 I-shaped etched slots. Due to the spacer of 1.5-cm-thick polycarbonate below the ground, the self-resonance of the slots provided a stable phase constant $\beta$ in the presence and absence of the phantom.

The LWA was fed from one side with a 50 Ω coaxial cable soldered to the microstrip line. The second port of the LWA was loaded by a 50 Ω high power resistor. In the prototype, neither a matching network nor a balun had to be used. In contrast, the dipole required a lumped-element network to be matched to 50 Ω and a ceramic balun placed on the feeding coaxial line.

For all measurements, a pelvis-shaped homogeneous phantom with similar sizes and electrical properties as in the numerical simulations was used. S-parameter measurements of the LWA and dipole antenna were measured using a Copper Mountain TR1300/1 2-Port vector network analyzer.

**MRI characterization**. Measurements of $\mathbf{B}_1^+$ and the temperature increment in a phantom, as well as MR imaging of a healthy volunteer, were performed on a 7 T Philips Achieva MRI-platform with eight transceiver channels (1 kW per each channel) available (Philips Healthcare, Best, The Netherlands) at the University Medical Center Utrecht, the Netherlands.

First, a homogeneous phantom was used to measure $\mathbf{B}_1^+$ patterns (Fig. 6a, b and e) created by the single LWA and dipole antenna located on the top of the phantom. The $\mathbf{B}_1^+$ maps were created using the Dual-TR method[41] (AFI) with the following scan parameters: Field of View: 231 mm × 400 mm × 140 mm, voxel size: 2.2 mm × 3.8 mm × 10 mm. TE/TR1/TR2: 2.2 μs/50 μs/250 μs.

The temperature maps were created using the proton resonance frequency shift method (Fig. 6c and d)[42]. The phantom was placed in the scanner room 1 h in advance to reach thermal equilibrium. Heating was provided by off-resonance (+100 kHz) block pulses with a power of 108 W on a 10% duty cycle, yielding an average power of 10.8 W. The following scan parameters were used: Field of View: 230 mm × 348 mm × 414 mm, voxel size: 3.6 mm × 4.8 mm × 6 mm, TE/TR: 10 μs/ 15 μs.

The in-vivo study of a healthy volunteer was approved by the local medical ethics committee, and informed consent was obtained from the subject.

In-vivo $\mathbf{B}_1^+$-maps was acquired using a similar method, that was used in the phantom measurement with following scan parameters: Field of View: 250 mm × 300 mm × 10 mm, voxel size: 1.25 mm × 1.25 mm × 10 mm. TE/TR1/TR2: 3 μs/50 μs/92 μs. $T_2$-weighted in-vivo body images (TR/TE = 2500 ms/90ms, 0.5 mm × 0.5 mm × 4mm, TSE-factor: 9) were obtained using the manufactured four-channel LWA array and the four-channel dipole array.

**Figure of merit for reactive near fields**. The relation between the reactive near fields and SAR, observed by our simulations, can be explained using Poynting's theorem written in a differential form with respect to the region of the phantom. Within this region, electromagnetic sources are absent, and only ohmic losses can be considered. In this case, for the time dependence taken in the form exp(j$\omega t$), the Poynting's theorem reads.

$$\nabla \cdot \mathbf{S} = -\frac{1}{2}\sigma|\mathbf{E}|^2 - j\omega\frac{1}{2}\left(\mu|\mathbf{H}|^2 - \varepsilon_0\varepsilon_r|\mathbf{E}|^2\right)$$
$$= -\frac{1}{2}\sigma|\mathbf{E}|^2 - j2\omega\Delta w, \tag{3}$$

where $\mathbf{S}$ is the Poynting vector, and $\Delta w$ is the difference of time-averaged magnetic and electric energy densities $\Delta w = w_{mag} - w_{el} = \frac{1}{4}\mu|\mathbf{H}|^2 - \frac{1}{4}\varepsilon_0\varepsilon_r|\mathbf{E}|^2$. In lossless regions permeated by propagating wave fields $\Delta w$ vanishes, whereas a significant difference occurs in regions of dominant reactive near fields, typically around resonating structures, due to the imposed spatial separation of electric and magnetic energy densities.

The spatial distribution of $\Delta w$ can thus be used as a qualitative measure to visualize the distinction between the domain of a coil's reactive near field and the adjacent region where (attenuated) propagating field solutions are assumed to emerge.

By considering equation (3), large values of $\Delta w$ are implicitly related to considerable conductive losses via involved electric fields respective electric currents. Large, spatially separated, but resonant electric and magnetic energy densities are interlinked by balancing current flows that are likely to contribute to enhanced SAR values.

Summarizing, the underlying relation of strong reactive near fields and SAR is given by the electric and magnetic field quantities as they have to fulfill Maxwell's equations and hence the Poynting's theorem in the context of an underlying boundary value problem. This results in spatial distributions of potentially high losses, which thus coincide with considerably large values for $\Delta w$, indicating large differences in the energy densities within this near field.

**Reporting summary**. Further information on research design is available in the Nature Research Reporting Summary linked to this article.

## Data availability
The data that support the presented results are available from the corresponding author upon request.

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

## Acknowledgements

Numerical and experimental studies of the leaky-wave antenna were supported by the Russian Foundation for Basic Research (Grant No. 19-29-10038). Numerical investigation of the different types of antenna elements was supported by the Russian Science Foundation (Project 19-75-10104). This project received funding from the European Union's Horizon 2020 research innovation program under grant agreement no. 736937. The authors thank Dr. Bart Steensma, Dr. Redha Abdeddaim, Prof. Constantin Simovski, Prof. Stefano Maci, Prof. Chris Collins, Ksenia Lezennikova, Ekaterina Brui and the team of the RF-coils lab at UMC Utrecht for their help and useful discussions.

## Author contributions

G.S. and J.T.S. are equally contributed to this work. G.S., S.G., A.R. and D.E. conceptualized the work. G.S. and J.T.S. performed electromagnetic simulations and data analysis. J.T.S. and A.R. carried out device fabrication and characterization. MRI experiments were preformed by A.J.R. and C.L. D.E. and S.G. provided supervision of project. All authors discussed the results and commented on the manuscript.

## Competing interests

The authors declare no competing interests.
