## [Peer Review File · Nature Communications]

Reviewers' comments:

Reviewer #1 (Remarks to the Author):

- Major claims of the paper

The authors present a non-resonant leaky wave antenna design, for which they optimize geometric parameters and employ it as a building block for a 4-channel prostate coil array for 7T MRI. The authors claim that with the new design, SAR can be considerably reduced whilst maintaining the same B1+ efficiency as a fractionated dipole antenna, which they use as a reference.

- Novelty/are the conclusions original?

Yes, to my knowledge there are no reports of leaky wave antennas applied as RF coils for MRI.

- Interest to others in the community and the wider field?

I found the paper very interesting to read and well structured. The results will definitely be of interest for the MR (UHF) community, since SAR efficiency is a major issue for coil design at UHF.

- Is further evidence required to strengthen the conclusions?

No

- Appropriateness and validity of statistical analysis

N/A

- Can the work be reproduced, given the level of detail provided?

Yes

- Comments

1. Would it be possible to change the size/geometry of the slots from one segment to the next in order to obtain a more symmetric pattern along the length of the conductor?

2. Can the authors comment on what effect an end load different from 50 Ohms (short/open/inductive/capacitive) would have? Could maybe reflections occur that would lead to leakage of the remaining RF power instead of dissipation in the end load?

3. I would personally remove the blue frame backgrounds of Figures 6, 7, and 9.

4. p.11/2nd paragraph: "The slot length $L_s=9.5$ cm was chosen...":

Somehow I cannot relate the numbers stated in the text to the values seen in Fig. 4. If I read Fig. 4b correctly, the theoretical radiation angle should be $\sim 20^\circ$ for $L_s=9.5$ cm, not 24.7° (which would be the reading for $L_s=12.4$ cm). For $\alpha \times p$ with phantom (Fig. 4e) I would read around 0.5 dB for $L_s=9.5$, assuming it corresponds to the 3rd red curve (seen from the right). Again, the curve with -1 dB would be for $L_s=12.4$ cm.

The value for beta seems to correspond to $\beta \times p$ of $\sim 55^\circ$, which in this case really is the value found on the 3rd red curve. Please clarify! Maybe it would be helpful to add a legend for the black-to-red lines in Fig.4, so one can relate the lines to a certain L_s . Also, $\alpha \times p$ should be labeled as negative numbers on the x-axis of Fig. 4e, 4f.

5. p.11/3rd paragraph: "... peak local SAR close to $a \times$ body surface ...":

replace "a" by "the"

6. p.11/4th paragraph: "... power radiated in to the phantom and absorbed by it calculated as ...":

What is S21 in this context? Is port 2 the matched load port? If so, please specify explicitly.

7. Fig. 6: add a cm-scale for size reference. If Fig 6c the value for B1+ in the ROI is missing. Also,

the measured B1+ map in Fig. 6c looks strange – it has a sharp transition along a horizontal line just below the ROI, can this be explained?

8. p.13/4th paragraph: "The measured S11...": The whole paragraph can be removed (has already been said on page 11)

9. p.14/line 4: replace "mm" by "cm".

10. p.14/line 5: "... (depths smaller than 5 cm)..." according to the orange region depicted in Fig. 7b, this should rather be ~2.8 cm. Also, please elaborate on why the measured B1+ is "saturated".

11. p.14/3rd paragraph: Please indicate how "simple phase shimming" was performed for the 4-channel array.

12. p.22/MRI characterization: Indicate the number of transmit channels and power/channel for the 7T scanner.

I recommend publication after a minor revision addressing my above comments.

- Elmar Laistler

Reviewer #2 (Remarks to the Author):

General Comments:

This paper presents a leaky-wave antenna (LWA) as a "coil" for MRI use. The overall performance of the LWA coil is better than the dipole that it is compared against, and this appears to be the first time that a LWA is used for this type of application. Therefore, I recommend publication. The paper seems to be a bit on the long side, however.

It is not clear how much the discussion about the stored energies in connection with Eq. (1) and Fig. 3 really add to the paper. The important aspect is what the SAR is near the coil, relative to the magnetic field in the region of interest (ROI). As the authors themselves say, this is not always directly correlated with the stored energy term in Fig. 3. I think it would be better to remove this part of the paper.

What would help a lot is for the authors to include a summary table that lists the comparison between the LWA coil and the dipole coil for a given input power (1W), comparing the field in the ROI, the peak SAR, and possibly the maximum temperature rise and the overall power being dissipated inside the body.

Specific Comments:

- 1) The paper does not seem to have an Introduction section.
- 2) It is not clear why the term "coil" is still used for the LWA and the dipole. They are not coils.
- 3) The acronym UHF clashes with the well-established acronym in RF terminology (Ultra High Frequency).

4) The final desired field is a circularly-polarized (CP) field. The LWA produces a linearly-polarized field. Are the eight LWAs phased to produce a CP field in the ROI? How does this effect the results and conclusions in the paper?

5) I assume that the red triangle in Fig. 6 is the feed, but this is not stated.

6) The discussion on p. 20 about the wave impedance does not really make sense to me. The fact that the wave impedance of the medium is 50 Ohms has nothing to do with what the input impedance of the antenna is. It is not necessary for the wave impedance (actually it should be called the intrinsic impedance) of the medium to be 50 ohms in order for the input impedance of the antenna to be 50 Ohms.

7) Have the authors also tried to optimize the length of the LWA? For a given amount of power left at the end (e.g, 10%), the length is still arbitrary.

8) Can the thickness of the polycarbonate layer be used to control the attenuation constant? How is the thickness chosen?

9) Equation (3) can be made exact by replacing the denominator with the magnitude of the beta vector, where the beta vector is the real part of the complex wavenumber vector, where the complex wavenumber vector has components $k_z = \beta - j(\alpha)$ and $k_y = \sqrt{k^2 - k_z^2}$, where k is the complex wavenumber of the medium.

10) Would the results be further improved if the LWA was fed symmetrically at both ends instead of only at one end?

Reviewer #3 (Remarks to the Author):

Review of 241679_0_art_file_4382194_q4v31n_convrt.pdf

• What are the major claims of the paper?

“A new excitation structure for UHF body imaging based on a leaky wave antenna element.”

• Are the claims novel? If not, please identify the major papers that compromise novelty. No, this topic is fairly broadly published by these authors and others, including some of the same figures and text of this manuscript. Three of a number of references are included here.

arXiv:2001.10410 [physics.app-ph]

Georgiy Solomakha, Carel van Leeuwen, Alexander Raaijmakers, Constantin Simovski, Alexander Popugaev, Redha Abdeddaim, Irina Melchakova, Stanislav Glybovski

Magn Reson Med. 1459-69, 81/2 2019.

<http://resolver.tudelft.nl/uuid:17eb2178-dd25-4ae5-b8ec-984229268768>

• Will the paper be of interest to others in the field? It's an interesting approach to RF coil design for MRI, but already available.

• Will the paper influence thinking in the field? Not likely more than what's already published.

• Are the claims convincing? If not, what further evidence is needed?

• The leaky wave antenna design has some potential merits for various antenna applications including for MRI. The effort to make these claims in this manuscript however was not conclusively convincing. The approach taken was very ambitious, comparing the leaky wave design to four other more commonly used RF coil elements. The comparisons however were very anecdotal in both design and experiment, much too limited to support the general conclusions drawn for the superior merits of the leaky wave design. Their claims were further buttressed by partial, often misleading or wholly incorrect arguments. Examples, are the claims for 7T not being clinical when in fact it is FDA approved and CE marked for clinical application by Siemens now. The major mode of clinical imaging which has been demonstrated at 7T uses separate transmit and receive coils.

This was not mentioned.

- Are there other experiments that would strengthen the paper further? How much would they improve it, and how difficult are they likely to be?

Yes, given this coil element is to revolutionize MRI, it would have been nice to see a recognizable MR image, for starters.

- Are the claims appropriately discussed in the context of previous literature? No. Significant literature was not referenced such as large body of work over two decades on this topic from the U. of Minnesota, Oxford and others. Here's a starter sampler:

Parallel transceiver for nuclear magnetic resonance system, US Patent 6,969,992 (and others)

Snyder CJ, DelaBarre L, Metzger GJ, van de Moortele PF, Akgun C, Ugurbil K, Vaughan JT. Initial results of cardiac imaging at 7 tesla. *Magnetic Resonance in Medicine*. 2009;61(3):517-24. doi: 10.1002/mrm.21895. PubMed PMID: WOS:000263608300003.

Metzger GJ, Snyder C, Akgun C, Vaughan T, Ugurbil K, Van de Moortele PF. Local B-1(+) shimming for prostate imaging with transceiver arrays at 7T based on subject-dependent transmit phase measurements. *Magnetic Resonance in Medicine*. 2008;59(2):396-409. doi: 10.1002/mrm.21476. PubMed PMID: WOS:000252901700019.

- If the manuscript is unacceptable in its present form, does the study seem sufficiently promising that the authors should be encouraged to consider a resubmission in the future? In the light of the work already published, no. If it were a consideration, a practical (clinical) demonstration of the clear advantages of this coil and how it will revolutionize high field MR imaging (with statistically conclusive results as opposed to handwaving) might work. This would be new and hasn't been convincingly shown.

- Is the manuscript clearly written? If not, how could it be made more accessible? No. The authors have attempted to show, do, claim, explain too much in one manuscript. The result is a big basketful and anecdotal bits leaving us with no convincing conclusion.

- Could the manuscript be shortened to aid communication of the most important findings? Yes. If this stayed on target with convincing us that a new MR coil could be explained, designed, built and demonstrated for its intended application, it might be more conclusively focused.

- Have the authors done themselves justice without overselling their claims? No, to the contrary. To distinguish their claims from the field, they have selectively chosen often only qualitatively explained advantages of their design while not discussing its disadvantages, or the relative advantages of other designs currently in use.

- Have they been fair in their treatment of previous literature?

No, per above description.

- Have they provided sufficient methodological detail that the experiments could be reproduced? Not fully. For example, though low SAR is integral to the title and claims of this work, the reason for this concern is barely mentioned. SAR can result in excessive heating, though SAR alone is not sufficient for determining either the location or the degree of heating. The one temperature measurement mentioned was nearly an order of magnitude more accurate and precise than can be made by the PRF method used as reported in the literature. Page 14/26 INTERNATIONAL JOURNAL OF HYPERTHERMIA 2018, VOL. 34, NO. 8, 1381–1389.

- Is the statistical analysis of the data sound?

No, everything is one of, anecdotal.

- Should the authors be asked to provide further data or methodological information to help others replicate their work? (Such data might include source code for modelling studies, detailed protocols or mathematical derivations).

If a rewrite were warranted, more detailed methods description would be helpful.

- Are there any special ethical concerns arising from the use of animals or human subjects? No.

Reviewers comments and comments from the authors

Reviewer 1 (Remarks to the Author)

Major claims of the paper

The authors present a non-resonant leaky wave antenna design, for which they optimize geometric parameters and they employ it as a building block for a 4-channel prostate coil array for 7T MRI. The authors claim that with the new design, SAR can be considerably reduced whilst maintaining the same B1+ efficiency as a fractionated dipole antenna, which they use as a reference.

Novelty/are the conclusions original?

Yes, to my knowledge there are no reports of leaky wave antennas applied as RF coils for MRI.

Interest to others in the community and the wider field?

I found the paper very interesting to read and well structured. The results will definitely be of interest for the MR (UHF) community, since SAR efficiency is a major issue for coil design at UHF.

Is further evidence required to strengthen the conclusions?

No

Appropriateness and validity of statistical analysis

N/A

Can the work be reproduced, given the level of detail provided?

Yes

Comments

1. Would it be possible to change the size/geometry of the slots from one segment to the next in order to obtain a more symmetric pattern along the length of the conductor?

AUTHORS: Yes, it is possible. In fact, one could introduce not just a linear but another shape of the spatial phase variation. However, to change the slots and by that their dispersive behavior from one unit cell to the next one the Bloch impedance should be also carefully controlled to avoid internal reflections. This is possible to do, but for the price of narrower bandwidth. In our approach we aimed for the most broadband behavior that corresponds to the low SAR, so we did not discuss this possibility.

2. Can the authors comment on what effect an end load different from 50 Ohms (short / open / inductive / capacitive) would have? Could maybe reflections occur that would lead to leakage of the

remaining RF power instead of dissipation in the end load?

AUTHORS: A strong reflective load (such as the given examples) will lead to standing waves inside the coil, or more specifically along the transmission line. This would result in a resonant coil instead of a desirable non-resonant one. During the design procedure, we tested several load options and all of them lead to lower B_1 -fields in the ROI and higher SAR levels.

To clarify this aspect, we added the following sentence to the paper: "To avoid reflections from the load, matched loads or power recycling networks [?] are connected to the end of the transmission line. Usually, LWAs are designed to radiate more than 80% of its guided input power. In this case, the terminal load has a negligible influence on the LWA performance. However, to avoid standing waves, the end load should be matched to the line."

3. *I would personally remove the blue frame backgrounds of Figures 6, 7, and 9.*

AUTHORS: Thank you for your feedback on the appearance of these figures. We prefer this style, but would be ready to remove the frames if requested by a journal.

4. *p.11/2nd paragraph: "The slot length $L_s=9.5$ cm was chosen...": Somehow I cannot relate the numbers stated in the text to the values seen in Fig. 4. If I read Fig. 4b correctly, the theoretical radiation angle should be 20 deg for $L_s=9.5$ cm, not 24.7 deg (which would be the reading for $L_s=12.4$ cm). For α x p with phantom (Fig. 4e) I would read around 0.5 dB for $L_s=9.5$, assuming it corresponds to the 3rd red curve (seen from the right). Again, the curve with -1 dB would be for $L_s=12.4$ cm. The value for beta seems to correspond to beta * p of 55 deg, which in this case really is the value found on the 3rd red curve. Please clarify! Maybe it would be helpful to add a legend for the black-to-red lines in Fig.4, so one can relate the lines to a certain L_s . Also, α * p should be labeled as negative numbers on the x-axis of Fig. 4e, 4f.*

AUTHORS: Thank you for this correction. This was a misprint that we overlooked in the 2nd paragraph of p.11 and we corrected the perimeter of a slot to be 12.4 cm. The curves for the chosen optimal length are correspondingly the ones indicated boldly in Fig.4.

Addressing your comments last sentence, we corrected the axis labels of the loss factor plots of Figure 4 to be insertion loss, cause this is truly the visualized value and therefore the axis is correctly positive. However, to clarify we updated the figures caption.

5. *p.11/3rd paragraph: "... peak local SAR close to *a* body surface ...": replace "a" by "the"*

AUTHORS: We corrected it.

6. *p.11/4th paragraph: "... power radiated in to the phantom and absorbed by it calculated as ...": What is S_{21} in this context? Is port 2 the matched load port? If so, please specify explicitly.*

AUTHORS: Yes, in the simulations we used two 50-Ohm ports: one at the beginning and the other at the end of the coil. We changed the text accordingly "The value ν means the part of the input power, which is neither reflected at the feed nor consumed by the end load. Generally, it includes power going to the phantom as well as losses due to radiation in free space and dissipation losses inside the coil."

7. Fig. 6: add a cm-scale for size reference. If Fig 6c the value for $B1+$ in the ROI is missing. Also, the measured $B1+$ map in Fig. 6c looks strange – it has a sharp transition along a horizontal line just below the ROI, can this be explained?

AUTHORS: We added the cm-scale and the missing value. The sharp transition has happened due to the presence of a technical hole in the pelvis-shaped phantom used in the measurement, in which typically a pick-up probe is installed during on-bench measurements. Also, below the ROI the level of SNR is too real to map B_1^+ of one element with sufficient accuracy. These aspects were explained in the text as follows: "The artifacts visible in the measured figures below the ROI, are due to a technological hole in the phantom and insufficient signal-to-noise ratio for B_1^+ mapping".

8. p.13/4th paragraph: "The measured $S11\dots$ ": The whole paragraph can be removed (has already been said on page 11)

AUTHORS: Thanks for an advice, we agree that this part was excess. We reduce this part and combine this with the other paragraph.

9. p.14/line 4: replace "mm" by "cm".

AUTHORS: We corrected it.

10. p.14/line 5: "... (depths smaller than 5 cm)..." according to the orange region depicted in Fig. 7b, this should rather be 2.8 cm. Also, please elaborate on why the measured $B1+$ is "saturated".

AUTHORS: We corrected this misprint, thank you. The saturation of the $B1$ level was due to the restrictions of the double TR method' dynamic range. We explained our choice of the comparison depth range in the text as follows: "The field level changes dramatically from the surface to the center of the phantom due to attenuation. Therefore to carefully compare the B_1^+ levels close to the ROI within the dynamic range of the B_1^+ mapping method, we had to over-saturate the field level measured at depths smaller than 3 cm."

11. p.14/3rd paragraph: Please indicate how "simple phase shimming" was performed for the 4-channel array.

AUTHORS: We explained this approach in the Methods section as follows: "Phase-only static RF

shimming, based on B_1^+ phases induced by each of four array elements [?], was applied to maximize B_1^+ in the prostate region. First, the B_1^+ in the ROI was exported to Matlab (Natick, Massachusetts, USA) for each channel, whereafter, the individual phases of excitation were determined and applied with the same magnitude.”

12. p.22/MRI characterization: Indicate the number of transmit channels and power/channel for the 7T scanner.

AUTHORS: The scanner Tx system allowed eight-channel configuration, while we used only four channels for simplicity. We gave this information in the Methods section as follows: ”Measurements of B_1^+ and the temperature increment in a phantom, as well as MR imaging of a healthy volunteer, were performed on a 7 T Philips Achieva MRI-platform with eight transceive channels (1 kW per each channel) available (Philips Healthcare, Best, The Netherlands) at the University Medical Center Utrecht, the Netherlands.”

Other comments

I recommend publication after a minor revision addressing my above comments. - Elmar Laistler

Reviewer 2 (Remarks to the Author)

General Comments

This paper presents a leaky-wave antenna (LWA) as a “coil” for MRI use. The overall performance of the LWA coil is better than the dipole that it is compared against, and this appears to be the first time that a LWA is used for this type of application. Therefore, I recommend publication. The paper seems to be a bit on the long side, however.

AUTHORS: Thank you for the positive reply. We worked carefully on the entire text to make the discussion more compressed and clear. Instead, more direct comparison data was added including *in-vivo* comparison.

It is not clear how much the discussion about the stored energies in connection with Eq. (1) and Fig. 3 really add to the paper. The important aspect is what the SAR is near the coil, relative to the magnetic field in the region of interest (ROI). As the authors themselves say, this is not always directly correlated with the stored energy term in Fig. 3. I think it would be better to remove this part of the paper.

AUTHORS: We partially agree with the Reviewer’s point, so we remove the most of this part from the main part of the text, leaving a more compressed paragraph about it in the Methods session. However, in the new version, we decided to keep using the energy difference figure of merit. In

fact, our idea was to relate the peak local SAR close to a coil with the reactive field produced. In contrast to the radiative field, in which E and H components are just proportional, in reactive field the electric component can locally grow producing SAR hotspots. However, mathematically the reactive field in a highly conductive medium is not determined in the sense it is known for antennas in free space. Therefore, to introduce the value ξ , which is related to spatial separation of E- and H-fields was the only way to distinguish the reactive and radiative fields. Using this quantity we can show that the proposed LWA coil element is the least reactive, then the dipole and then all other coils. So to make it closer to the aim of the paper, we kept in the main text only a shortened and clearer description of this quality and its physical meaning along with its plot in Figure 3 for different coils.

What would help a lot is for the authors to include a summary table that lists the comparison between the LWA coil and the dipole coil for a given input power (1W), comparing the field in the ROI, the peak SAR, and possibly the maximum temperature rise and the overall power being dissipated inside the body.

AUTHORS: We included two tables with that data. The first one (Table 1) is related to the operation of the LWA coil element compared to four other coil types shown in Fig. 2, simulated as single elements on a homogeneous phantom. The second one (Table 2) is to support the SAR benefit of LWA coil element by a more careful comparison of the same coil types in the four-element array configurations, calculated used voxel-based anatomical body models. We hope that thanks to this additional data calculated since the previous version, our conclusions are better supported.

Specific Comments

1) *The paper does not seem to have an Introduction section.*

AUTHORS: We structured this paper according to the recommended template Nature Communications, and considered other papers published in this journal, where there was no Introduction section. However, we have an introduction text which starts right after the abstract and goes until the Results section.

2) *It is not clear why the term "coil" is still used for the LWA and the dipole. They are not coils.*

AUTHORS: "Coil" is a commonly used term for on-body transmit and receive antennas in the MRI community no matter which implementation they have. Such antennas are called "coils" historically because first such devices were coils. However, we agree that in our case the only acceptable terminology is "array coil of leaky-wave antennas", while a single element should be called "antenna" or "LWA coil element". We will change it in the text except for the running head,

where we leave "leaky-wave coil" for shortness.

3) *The acronym UHF clashes with the well-established acronym in RF terminology (Ultra High Frequency).*

AUTHORS: We agree, this acronym is not used in the new version

4) *The final desired field is a circularly-polarized (CP) field. The LWA produces a linearly-polarized field. Are the eight LWAs phased to produce a CP field in the ROI? How does this effect the results and conclusions in the paper?*

AUTHORS: Yes, indeed, in case of more than one driven LWA the array is phased in order to create a CP field in the ROI. However, the fields of the single LWAs are able to be linearly superpositioned therefore the SNR increases (in a first-order approximation) linearly with the number of coil-elements. In order to make fair comparisons between array elements of different types (e.g. leaky-wave elements vs. dipoles), the number of elements as well as the positioning of the elements should be the same.

5) *I assume that the red triangle in Fig. 6 is the feed, but this is not stated.*

AUTHORS: We added this inscription to Figure 6 and to its caption.

6) *The discussion on p. 20 about the wave impedance does not really make sense to me. The fact that the wave impedance of the medium is 50 Ohms has nothing to do with what the input impedance of the antenna is. It is not necessary for the wave impedance (actually it should be called the intrinsic impedance) of the medium to be 50 ohms in order for the input impedance of the antenna to be 50 Ohms.*

AUTHORS: We agree with the Reviewer that the input impedance and the wave impedance of the surrounding medium are not the same things, but, for sure, these two characteristics are related. The same antenna would have different radiation impedance in media with different permittivity. But we also agree that we do not show enough results to relate in 50 Ohm input impedance of the on-body leaky-wave antenna with 50 Ohm wave impedance of the medium with the permittivity of around 50 Ohm. So this claim was, indeed, not supported and we removed it. Instead it would be correct to say that our antenna is self-matched because there is no reflection at the end and all the power is radiated while traveling-wave propagation in the microstrip. We keep the following explanation in the new text: "As long as the propagating wave loses most of its power in the form of radiation before reaching the end of the line, the antenna is non-resonant, and its input impedance is relatively stable with frequency. To avoid reflections of the load, matched loads or power recycling networks[37] are connected to the end of the line. Usually LWAs are designed

to radiate more than 80 % of power. In this case the end load does not play a considerable role. However, to avoid standing waves, the end load should be matched to the line.

7) *Have the authors also tried to optimize the length of the LWA? For a given amount of power left at the end (e.g, 10%), the length is still arbitrary.*

AUTHORS: The chosen length of the LWA actually was chosen to radiate at least 60% of power into a phantom. The leakage factor is determined by the perimeter of the slots. And since we limited the width of the antenna to be 8.5 cm to contain in principle 8 elements in a body array, the leakage factor was limited from above by the limited possible perimeter of slots. Therefore, the length could not be made considerably smaller. Otherwise the losses in the end load would increase and efficiency would degrade. On the other hand, one cannot easily place antennas longer than 40 cm around a human pelvis without large air gaps, which are not desirable. Therefore, we preferred 40 cm. The corresponding explanation was added to the text: "Therefore, we limited its length by 40 cm and width of 8.5 cm, to be able to place eight such coils around a human pelvis. In fact, we used the maximum available length to maximize the propagation length of the exponentially decaying wave inside the microstrip line and radiate as much applied power as possible before reaching the end load. With the same aim the slot geometry was configured"

8) *Can the thickness of the polycarbonate layer be used to control the attenuation constant? How is the thickness chosen?*

AUTHORS: Yes, of course. For the thinner polycarbonate it is easier to obtain larger attenuation constant. However, shortening is also harmful for the bandwidth and therefore - the amount of reactive fields. As a result for a thinner spacer the level of peak local SAR would increase. We addressed this aspect in the new text as follows: "For the given length of the coil, the chosen thickness results from the optimal trade-off between B_1^+ in the ROI and the peak local SAR. In fact, for thicknesses less than 1.5 cm the level of SAR of the leaky-wave coil element is worse than compared to a dipole element with equal levels of B_1^+ in the ROI."

9) *Equation (3) can be made exact by replacing the denominator with the magnitude of the beta vector, where the beta vector is the real part of the complex wavenumber vector, where the complex wavenumber vector has components $k_x = \beta - j(\alpha)$ and $k_y = \sqrt{k^2 - k_z^2}$, where k is the complex wavenumber of the medium.*

AUTHORS: We have adjusted the formula and also updated the text with regard to this change. Due to this change, the estimated angle is marginally smaller and the angle can still only be used as an approximate value, since the short leaky-wave structure makes it difficult to determine a

radiation angle anyway.

10) *Would the results be further improved if the LWA was fed symmetrically at both ends instead of only at one end?*

AUTHORS: Yes, we agree that it is possible to use that approach to make field pattern more symmetrical. Moreover, one could improve power efficiency. However, since a standing wave is produced inside the antenna from two waves travelling towards each other, SAR efficiency is negatively affected.

Reviewer 3 (Remarks to the Author)

What are the major claims of the paper?

“A new excitation structure for UHF body imaging based on a leaky wave antenna element.”

Are the claims novel? If not, please identify the major papers that compromise novelty.

No, this topic is fairly broadly published by these authors and others, including some of the same figures and text of this manuscript. Three of a number of references are included here.

*arXiv:2001.10410 [physics.app-ph] Georgiy Solomakha, Carel van Leeuwen, Alexander Raaijmakers, Constantin Simovski, Alexander Popugayev, Redha Abdeddaim, Irina Melchakova, Stanislav Glybovski *Magn Reson Med.* 1459-69, 81/2 2019. <http://resolver.tudelft.nl/uuid:17eb2178-dd25-4ae5-b8ec-984229268768>*

AUTHORS: We strongly disagree with the claim by Reviewer 3 regarding the novelty. Indeed, leaky-wave antennas in the microwave range is a widely elaborated topic. Co-authors of the present paper also contributed previously into this topic. We give an appropriate outlook for leaky-wave antennas and their types in the introduction text. Also we specially mentioned several examples of leaky-wave applicators, i.e. leaky-wave antennas that were used at microwaves not to radiate into free space, but into a dielectric medium. Among them were ground penetrating radar antennas and lens feeding antennas. However, we have no doubts and we checked carefully again that a leaky-wave antenna was first time demonstrated in MRI in this our paper. Moreover, it is the first to our knowledge non-resonant surface coil for MRI.

The references proposed by the Reviewer 3 are not related to the proposed idea. Thus in [<http://resolver.tudelft.nl/uuid:17eb2178-dd25-4ae5-b8ec-984229268768>] an application of leaky-wave antennas for microwave imaging is considered. The results of this work have no relation to MRI, and even microwave imaging is opposed to MRI. However, we added this reference to our paper to mention one more microwave applications of leaky-wave antennas.

Next, [arXiv:2001.10410 [physics.app-ph]] is just a preprint of the present paper, which according to the journal policies we can store at ArXiv at any time of the review process to claim the novelty of our idea.

Finally, [Georgiy Solomakha, Carel van Leeuwen, Alexander Raaijmakers, Constantin Simovski, Alexander Popugaev, Redha Abdeddaim, Irina Melchakova, Stanislav Glybovski Magn Reson Med. 1459-69, 81/2 2019] is our previous paper, which was also about SAR reduction. But the method we proposed there was completely different. We proposed to combine two parallel dipole antennas instead of one to increase bandwidth and reduce SAR. This paper had no relation to leaky-wave antennas and travelling-wave MRI.

So we cannot accept the claims of the reviewer against the novelty of our idea of a non-resonant surface coil based of LWA.

Will the paper be of interest to others in the field?

It's an interesting approach to RF coil design for MRI, but already available.

AUTHORS: We disagree. See above for details..

Will the paper influence thinking in the field?

Not likely more than whats already published.

AUTHORS: We disagree. See above for details.

Are the claims convincing? If not, what further evidence is needed?

The leaky wave antenna design has some potential merits for various antenna applications including for MRI. The effort to make these claims in this manuscript however was not conclusively convincing. The approach taken was very ambitious, comparing the leaky wave design to four other more commonly used RF coil elements. The comparisons however were very anecdotal in both design and experiment, much to limited to support the general conclusions drawn for the superior merits of the leaky wave design. Their claims were further buttressed by partial, often misleading or wholly incorrect arguments. Examples, are the claims for 7T not being clinical when in fact it is FDA approved and CE marked for clinical application by Siemens now. The major mode of clinical imaging which has been demonstrated at 7T uses separate transmit and receive coils. This was not mentioned.

AUTHORS: We agree partially. In the new version of the paper we removed all mentions of potential clinical usage of LWA coil element. For now, only head and extremities imaging at 7 Tesla approved for clinical usage [<https://www.fda.gov/news-events/press-announcements/fda-clears-first-7t-magnetic-resonance-imaging-device>], body imaging is still developing research instrument [Gruher B, Froeling M, Leiner T, Klomp DWJ. RF coils: A practical guide for nonphysicists. Journal

of Magnetic Resonance Imaging. 2018;48(3):590–604.].

As to the comparison of the leaky-wave antenna to other coils, our main goal was to justify the relation between resonant properties (Q-factor and reactive fields) with maximum local SAR. In simple words: there is a trend described in the literature to more radiative and low-Q coils for body imaging at 7T. We needed to show it because our aim was to continue this trend by introducing a completely non-resonant coil, which has even lower SAR than a fractionated dipole. At this point we compare the dipole and the leaky-wave antenna to each other in details. In the new version we added comparison of SAR by simulation on two body models and also in-vivo comparison. In summary, we do not aim to overcome every existing coil, but we are sure that for SAR we can compete with the dipole. Also, our leaky-wave antenna is matched automatically when placed onto a body and this is a unique property. We tried to clarify our claims throughout the new version of the text.

Are there other experiments that would strengthen the paper further? How much would they improve it, and how difficult are they likely to be?

Yes, given this coil element is to revolutionize MRI, it would have been nice to see a recognizable MR image, for starters.

AUTHORS: In the new version we provided a careful comparison of body images in four-channel configuration of dipoles and leaky-wave antennas (in bot case shimming for the prostate). Similar quality images have been provided previously when other coils for prostate imaging at 7T were introduced. For example [Fractionated dipole introduced <https://doi.org/10.1002/mrm.25596>]. We stress that the benefit we propose is not is better efficiency and imaging but in lower SAR and reactive fields as well as in self-matching. So we believe the examples of body images we compare are enough to confirm that imaging quality is comparable to dipoles, and therefore support the applicability of a new physical mechanism of excitation in MRI.

Are the claims appropriately discussed in the context of previous literature? *No. Significant literature was not referenced such as large body of work over two decades on this topic from the U. of Minnesota, Oxford and others. Here's a starter sampler: Parallel transceiver for nuclear magnetic resonance system, US Patent 6,969,992 (and others) Snyder CJ, DelaBarre L, Metzger GJ, van de Moortele PF, Akgun C, Ugurbil K, Vaughan JT. Initial results of cardiac imaging at 7 tesla. Magnetic Resonance in Medicine. 2009;61(3):517-24. doi: 10.1002/mrm.21895. PubMed PMID: WOS:000263608300003. Metzger GJ, Snyder C, Akgun C, Vaughan T, Ugurbil K, Van de Moortele PF. Local B-1(+) shimming for prostate imaging with transceiver arrays at*

7T based on subject-dependent transmit phase measurements. *Magnetic Resonance in Medicine*. 2008;59(2):396-409. doi: 10.1002/mrm.21476. PubMed PMID: WOS:000252901700019.

AUTHORS: We appreciate, we added that references and comments about work of Minnesota group. This works have great importance in this field of RF coils and body UHF imaging, but this do not affect our conclusion.

If the manuscript is unacceptable in its present form, does the study seem sufficiently promising that the authors should be encouraged to consider a resubmission in the future?

In the light of the work already published, no. If it were a consideration, a practical (clinical) demonstration of the clear advantages of this coil and how it will revolutionize high field MR imaging (with statistically conclusive results as opposed to handwaving) might work. This would be new and hasn't been convincingly shown.

AUTHORS: As we indicated previously, this is not fully correct. First of at all we do not aim for clinical applications, since UHF body imaging is still purely research instrument. We are no claiming that we will "revolutionize high field MR imaging", we just want to report new physical mechanism of RF-excitation in MRI that will be useful in future development of body imaging approaches at 7T due to lower SAR and automatic impedance matching.. We agree with the constructive criticism by the Reviewer concerning our results and their comparison made. In order to improve it and make our results as statistically conclusive as possible, in the new version we added comparative simulations of four-channels arrays made of dipole and of leaky-wave antennas. We provide SAR and efficiency comparison on two body anatomical models with different BMI. Moreover we organised another measurements, where we experimentally compare B_1^+ maps and body images on a healthy volunteer. In this case, the same four-channel configuration was used. The new results confirm that the efficiency is comparable but SAR is lower. We agree with the reviewer that in-vivo comparison should be also statistical and made on several healthy volunteers. But currently such studies are impossible to organise for our team due to the pandemic situation.

Is the manuscript clearly written? If not, how could it be made more accessible?

No. The authors have attempted to show, do, claim, explain too much in one manuscript. The result is a big basketful and anecdotal bits leaving us with no convincing conclusion.

AUTHORS: We agree with the criticism and put our efforts to clarify and shorted the discussion where applicable. Moreover, we improved the results representation, added more important comparison examples including in-vivo, and also revised our conclusions.

Could the manuscript be shortened to aid communication of the most important findings?

Yes. If this stayed on target with convincing us that a new MR coil could be explained, designed, built and demonstrated for its intended application, it might be more conclusively focused.

AUTHORS: We agree. Please see above for details.

Have the authors done themselves justice without overselling their claims?

No, to the contrary. To distinguish their claims from the field, they have selectively chosen often only qualitatively explained advantages of their design while not discussing its disadvantages, or the relative advantages of other designs currently in use.

AUTHORS: We partially agree with that point as well and improved the discussion accordingly. In particular, we added a paragraph describing disadvantages. Thus the proposed antenna is relatively bulky and more sensitive to the presence of air gaps between the spacer and a subject as compared to the dipole. As for the selective approach, we do not agree. We compared our coil with the fractionated dipole in the way it is accepted in the literature without any selection in our favor. As for the comparison with our coils, it was mostly done for demonstrating the relationship between Q-factor and peak local SAR. However, as requested, we improved that part either by adding simulations also on an anatomical body model.

Have they been fair in their treatment of previous literature?

No, per above description.

AUTHORS: Please see above.

Have they provided sufficient methodological detail that the experiments could be reproduced?

Not fully. For example, though low SAR is integral to the title and claims of this work, the reason for this concern is barely mentioned. SAR can result in excessive heating, though SAR alone is not sufficient for determining either the location or the degree of heating. The one temperature measurement mentioned was nearly an order of magnitude more accurate and precise than can be made by the PRF method used as reported in the literature. Page 14/26 INTERNATIONAL JOURNAL OF HYPERTHERMIA 2018, VOL. 34, NO. 8, 1381–1389.

AUTHORS: We agree, that SAR alone is not sufficient for determining either the location or the degree of heating, but remains the main safety factor of RF part of MRI according to recommendations of IEC (<https://webstore.iec.ch/publication/28673>) and FDA (<https://www.fda.gov/radiation-emitting-products/mri-magnetic-resonance-imaging/information-industry>). Therefore, we believe that for

the goals of our work (demonstration of lower E-fields created by a non-resonant surface coil), investigation of SAR is sufficient.

Heating in our experiments was normalized to accepted power (16 Watts in our case), like we did with results of SAR simulations. To clarify this quantity normalization, that may be indeed, misleading, we added the following changes to the paper: - in caption of Figure 6: " Simulated and measured field distributions for 1 W of accepted input transmit power...local SAR and ΔT in the top plane (XZ) for LWA (e,g) and dipole (f,h)."

- in the text: "Resulting temperature distributions, normalized to the accepted power, are presented in Figure 6 (g) and (h)/

Is the statistical analysis of the data sound?

No, everything is one of, anecdotal.

AUTHORS: Major claims of this paper is related to research 7 T body imaging field, not for clinical usage. Moreover, in the current pandemic situation and limited access to scanners, we don't have resources to provide statistical data *in-vivo*. However, to address this point as mush as we can, and to additionally check SAR reduction we perform B_1^+ and SAR simulations with two different body models (examples of simulation results made are given in Figures 1 and 2 of this response letter). The corresponding calculated data is summarized in the new version of the paper in a Table 2. Simulations of B_1^+ and SAR of four-channel loop, stripline and slot arrays on Duke anatomical model also were performed. Also, we performed a new set of *in-vivo* measurements B_1^+ both for a leaky-wave and dipole in four channel to prove results of numerical simulations, and finally provided a comparison of images.

Should the authors be asked to provide further data or methodological information to help others replicate their work? (Such data might include source code for modelling studies, detailed protocols or mathematical derivations).

If a rewrite were warranted, more detailed methods description would be helpful, while the entire text was revised and clarified.

AUTHORS: We agree and added more information to the methods part, while the entire text was revised and clarified.

Are there any special ethical concerns arising from the use of animals or human subjects?

No.

Figure 1: Simulated B_1^+ and SAR_{10g} for the LWA and dipole four-channel coil array on Duke and Gustav anatomical model.

Figure 2: Simulated B_1^+ and SAR_{10g} for the LWA, dipole, stripline, slot and loop four-channel coil array on Duke anatomical model.

REVIEWERS' COMMENTS

Reviewer #1 (Remarks to the Author):

Thank you for addressing my comments, the manuscript has considerably improved and I recommend publication as is.

Reviewer #3 (Remarks to the Author):

Reviewer 3:

The authors present a leaky wave antenna approach for ultra-high field MRI. This review follows the author's revision of their original manuscript.

The authors were thoroughly responsive, and in most cases acceptably so. The other two reviewers were both expert and favorably disposed to recommending this manuscript for publication with minor revision. Given these two factors, this reviewer will follow the consensus here save for his first and strongest criticism, that this work is already published. If Nature is OK with publishing work already in publication in other journals, then I concede on this point and agree to support publication of this otherwise improved manuscript.

While this reviewer does not dispute the author's original and creative contribution to the application of LWAs to MRI, and does recognize its potential value to the field, Reviewer 3's major criticism was and still is that this work has already been published. Since this criticism was emphatically denied by the authors, I copy the title, authorship and abstract of one such recent publication. To this reviewer it looks very much the same as this manuscript under review. But if it's not, please accept my concession.

A Self-Matched Leaky-Wave Antenna for Ultrahigh-Field MRI with Low SAR

G. Solomakha, J. T. Svejda, C. van Leeuwen, A. Rennings, A. J. Raaijmakers, S. Glybovski, D. Erni

The technology of magnetic resonance imaging is developing towards higher magnetic fields to improve resolution and contrast. However, whole-body imaging at 7 T or even higher fields remains challenging due to wave interference, tissue inhomogeneities and high RF power deposition. Nowadays, proper RF excitation of a human body in prostate and cardiac MRI is only possible to achieve by using phased arrays of antennas attached to the body (so-called surface coils). Due to safety concerns, the design of such coils aims to minimize the local specific absorption rate (SAR) keeping the highest possible RF signal in the region of interest. All previously demonstrated approaches were based on resonant structures such as e. g. dipoles, capacitively-loaded loops, TEM-line sections. In this study, we show that there is a better compromise between the transmit signal and the local SAR using non-resonant surface coils due to weaker RF near fields in the close proximity of their conductors. With this aim, we propose and experimentally demonstrate a first leaky-wave surface coil implemented as a periodically-slotted microstrip transmission line. Due to its non-resonant radiation, the proposed coil induces only slightly over half the peak local SAR compared to a state-of-the-art dipole coil, but has the same transmit efficiency in prostate imaging at 7 T. Unlike other coils, the leaky-wave coil intrinsically matches its input impedance to the averaged wave impedance of body tissues in a broad frequency range, which makes it very attractive for future clinical applications of 7 T MRI.

Comments: 26 pages, 9 figures

Subjects: Applied Physics (physics.app-ph)

Cite as: arXiv:2001.10410 [physics.app-ph]

(or arXiv:2001.10410v1 [physics.app-ph] for this version)

Bibliographic data

[Enable Bibex (What is Bibex?)]

Submission history

From: Georgiy Solomakha [view email]

[v1] Tue, 28 Jan 2020 15:27:25 UTC (8,164 KB)

Considering the remaining reviews and responses, This reviewer offers the following philosophical commentary. The MR field does need new and improved coil designs, and new designs including this one are always welcomed and encouraged. This review looks forward to seeing the work of this manuscript developed further in this regard; it does look like it may have some promising applications. This reviewer's main criticisms were publication protocol (above) and manuscript style (below). Comments below are general.

It's a common amateur's mistake that many of us never outgrow, to overclaim the importance of one's own work and overlook the work of others. This reviewer has witnessed 35 years of novel RF coil designs. "Novel coil design" has become cliché. An engineer builds a simple circuit based on a simple model. If they do few bench or phantom tests, that's a plus. If we see a convincing clinical demonstration or adaptation, that's a rarity and the Nature paper in this reviewer's opinion. But too frequently this circuit is claimed to be a significant new invention, given a sexy new name, and broadly claimed to be generally superior to everything that's come before, with factors of improvement, again based on thin and anecdotal evidence from the bench or theoretical model. References are commonly given as/for inferior designs. Less typical are the references of previous, same or similar contributions, and almost never is superior work acknowledged.

The better paper is a more focused report with specific and more solidly supported claims. eg. Based on these design criteria for this application I built this coil and report these results, period. Based on these promising findings, we plan to extend to these steps, a, b, c.

Most of these mistakes I know as my own,

Sincerely and good luck with this! I personally would like to see this developed further.

Tommy Vaughan